# DEVOLUTION—A method for phylogenetic reconstruction of aneuploid cancers based on multiregional genotyping data

Natalie Andersson [1✉], Subhayan Chattopadhyay[1], Anders Valind[1,2], Jenny Karlsson[1] & David Gisselsson [1,3,4]

Phylogenetic reconstruction of cancer cell populations remains challenging. There is a particular lack of tools that deconvolve clones based on copy number aberration analyses of multiple tumor biopsies separated in time and space from the same patient. This has hampered investigations of tumors rich in aneuploidy but few point mutations, as in many childhood cancers and high-risk adult cancer. Here, we present DEVOLUTION, an algorithm for subclonal deconvolution followed by phylogenetic reconstruction from bulk genotyping data. It integrates copy number and sequencing information across multiple tumor regions throughout the inference process, provided that the mutated clone fraction for each mutation is known. We validate DEVOLUTION on data from 56 pediatric tumors comprising 253 tumor biopsies and show a robust performance on simulations of bulk genotyping data. We also benchmark DEVOLUTION to similar bioinformatic tools using an external dataset. DEVOLUTION holds the potential to facilitate insights into the development, progression, and response to treatment, particularly in tumors with high burden of chromosomal copy number alterations.

[1] Division of Clinical Genetics, Department of Laboratory Medicine, Lund University, Lund, Sweden. [2] Department of Pediatrics, Skåne University Hospital, Lund, Sweden. [3] Division of Oncology-Pathology, Department of Clinical Sciences, Lund University, Lund, Sweden. [4] Clinical Genetics and Pathology, Laboratory Medicine, Lund University Hospital, Skåne Healthcare Region, Lund, Sweden. ✉email: natalie.andersson@med.lu.se

Neoplasms are a heterogenous group of diseases driven by Darwinian selection. Most cancers are presumed to originate from a single mutated cell, from which each mutation is conveyed to its daughter cells, that in turn can acquire additional aberrations, establishing subpopulations (subclones) of cells with diverse genetic compositions within the tumor[1]. This evolution of cancer cells is further shaped by genetic drift and selection pressure from the tumor microenvironment and oncological treatment[2–5]. Due to this process, many cancers exhibit vast intratumor heterogeneity (ITH) as well as intertumor heterogeneity between the primary tumor and its metastases[6–10]. Knowledge about how ITH emerges over time remains limited and multiple models have been proposed to explain it such as punctuated, neutral, linear, and branched evolution as well as a big bang model of tumor growth followed by neutral evolution[11–13]. By analyzing the genetic variation of the tumor spatially as well as temporally, mathematical methods can be employed in order to reconstruct its evolution, commonly in the form of a phylogenetic tree that links together distinct cancer cell subpopulations in an inferred temporal order. Such phylogenetic reconstructions can improve the understanding of tumorigenesis, progression to metastatic disease, and aid the development of novel therapeutic strategies[7,8,14].

One of the biggest challenges in phylogenetic analysis of bulk sample data from tumors is that the genetic analysis of each sample is conducted on millions of cells at once, usually constituting multiple subclones. The relative proportions of the subclones within each biopsy may also vary across the biopsied regions, stressing the need to integrate information from multiple biopsies separated in space to thoroughly assess the genetic profile of the tumor. Not addressing this may result in the prediction of illicit biological trajectories and so-called biopsy trees, not constituting true phylogenies[15]. Phylogenetic relationships should thus ideally be constructed based on the deconvolved clonal structure, i.e., one should infer which subclones are characterized by which alterations and, in addition, the ancestral order of these.

Even though single cell sequencing (SCS) has emerged as an important tool for temporal reconstruction that circumvents the issue of clonal deconvolution it is very costly to implement on the scales needed in the clinic and usually provides limited sequence coverage[16]. A more cost-effective alternative is to perform computerized deconvolution of bulk genotyping data derived from single nucleotide polymorphism array (SNP-array), targeted deep sequencing (TDS), whole exome sequencing (WES) or whole genome sequencing (WGS). Bulk genotyping yields a set of genetic alterations present in each biopsy along with information (e.g., log2 ratios, B allele frequencies, and variant allele frequencies) that together with allelic composition can be utilized to estimate the proportion of cells harboring each aberration in each biopsy, denoted the mutated clone fraction (MCF)[8,17–19]. Because MCF:s can be calculated for allelic/copy number imbalances and sequence mutations alike, it provides an excellent parameter for clonal deconvolution based on integrated data on these two types of genetic changes. However, most tools developed for computerized deconvolution of bulk genotyping data focuses solely on somatic point mutations, presumes a diploid background, lacks specific pipelines to handle intratumoral heterogeneity of copy number alterations (CNAs) encompassing chromosomal segments or whole chromosomes, and often assumes that the genetic alteration is present in all biopsies. In addition, they do not provide the possibility to infer phylogenetic trees solely on copy number aberration data from multiple biopsies separated in time and space[20–23]. Since many cancers are aneuploid to some degree[24], this is a serious shortcoming, especially for cancer types where aneuploidy is a common feature such as high-grade adult carcinomas, high-grade brain tumors, and many childhood

cancers[8,25,26]. Consequently, there is a particular need for tools capable to infer phylogenetic trees based on multiregional copy number data.

To fill this methodological gap, we introduce DEVOLUTION, an algorithm for subclonal deconvolution followed by phylogenetic reconstruction from bulk genome profiles including high-resolution copy number data (e.g., from SNP-array, WES or WGS) and sequencing information (e.g., from WES, WGS or TDS) separately or in unison. The deconvolution is based on à priori MCF-estimation of the individual aberrations in each sample and the algorithm systematically combines information from all available biopsies throughout the inference process to reconcile the most probable temporal evolution of the tumor by inferring an event matrix that is used to reconstruct phylogenetic trees. Importantly it can deduce evolutionary trajectories based on copy number data alone. In addition, predictions of the subclonal size and compositions across biopsies are visualized directly in the phylogenetic tree. DEVOLUTION provides an objective framework for creating event matrices and phylogenetic trees from bulk genotyping data, avoiding subjective bias compromising the validity of tree-to-tree comparisons (Supplementary Table 1).

To demonstrate DEVOLUTION's utility, the algorithm was evaluated using SNP-array data from 253 tumor regions from 56 pediatric cancers including neuroblastoma (NB), Wilms tumor (WT), and rhabdomyosarcoma (RMS)[17], comprising the most common extracranial, solid tumors in children. Additionally, extensive comparisons were made for WES-data alone, SNP-array data alone, and when using the two data sets conjointly for 18 of the tumors. The algorithm also showed a robust performance on simulated and publicly available multiregional bulk genotyping data. DEVOLUTION holds the potential to facilitate further insights into the development, progression, and response to treatment, particularly in tumors with a high burden of chromosomal copy number alterations.

## Results

**Overview of the algorithm workflow.** The algorithm operates on multiregional sampling data analyzed using whole genome profiling followed by MCF-computation (Fig. 1a–c). The input file is an $u \times v$ dimensional matrix, containing information about the $u$ genetic alterations detected in a tumor. For each alteration there are $v$ columns indicating the genetic position, alteration type as well as the proportion of cells in each biopsy harboring the alteration (Supplementary Table 2). For copy number aberrations, the matrix is subjected to an algorithm identifying all unique events across the samples while considering the uncertainty in the aberration breakpoint measurement (Supplementary Fig. 1). The DBSCAN (density-based spatial clustering of applications with noise) algorithm is then used to identify clusters of genetic alterations having similar cellular proportions across multiple samples, indicating that they might be reflecting a group of cells having an identical genetic profile (Fig. 1d)[27]. By identifying these clusters of genetic alterations, the computational load can be decreased and the unfolding of the subclonal composition aided.

DBSCAN prepares the data set to be subjected to the deconvolution algorithm which is employed to elucidate the temporal order of the clusters of mutations, deducing the subclones present across the biopsies. Information from multiple biopsies is integrated throughout this process to minimize the occurrence of parallel evolution (PLC) and back mutation contradictions (BMC). PLC in copy number data means that the same type of genetic alteration with the same genomic start- and endpoints in the chromosome appears independently in

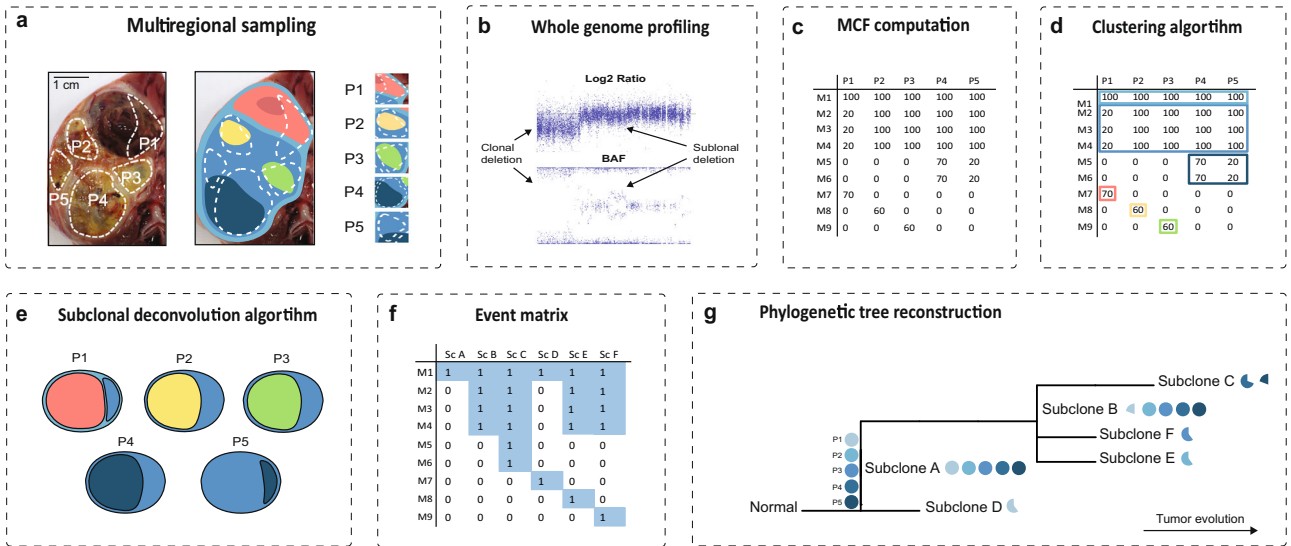

**Fig. 1 Overview of the methodological outline. a** An example of multiregional sampling to obtain biopsies P1-P5 from a Wilms tumor. The tumor is composed of several subclones with distinct genomic profiles, exemplified by the schematic genomic landscape in the rightmost panel where each color unifies cells with identical sets of genetic alterations. The photograph is adapted from a previous publication[14] and the colors do not represent true sets of genetic alterations. **b** Acquisition of genomic tumor data for each sample, exemplified by copy number analysis by SNP-array. **c** The whole genome profiling data can be used to compute the mutated clone fraction (MCF) illustrating the proportion of cells in each biopsy harboring a certain genetic alteration (M1-9 in the left column). **d** A clustering algorithm is employed to identify genetic alterations that seem to follow each other in size across samples. **e** A subclonal deconvolution algorithm determines the temporal order of these clusters by considering the information obtained throughout all samples while minimizing the occurrence of parallel evolution and back mutations. **f** The proposed solution for the temporal order of subclones (Sc) is integrated into an event matrix. **g** This event matrix can be used to generate phylogenetic trees with either the maximum likelihood or the parsimony method. At the stem, all available biopsies for the patient are visualized as filled circles with the biopsy name (P1–P5). At the branches of the tree the subclones can be seen along with pie charts illustrating in which biopsies, and in what fraction of the tumor cells they appear.

different cells within the tumor. This is, in most cases, unlikely from a biological standpoint, unless the copy number alteration incorporates an entire chromosome or chromosome arm. BMC implicate genetic alterations that are gained and then lost further down the evolutionary history in the tree, which may be feasible scenarios for some types of genetic alterations, such as a gain of a whole chromosome that is later lost, but are less likely to occur for structural chromosomal aberrations and point mutations, and should never occur for loss of heterozygosity events[28].

In addition, the user can provide a matrix containing information about illicit orders of genetic events, that can be taken into consideration during the deconvolution (Fig. 1e). The deconvolution culminates in a suggestion of the most likely temporal order of all genetic aberrations, constituting the basis for the creation of an event matrix, illustrating the distribution of genetic alterations across subclones (Fig. 1f). Using this event matrix, the biological distance, representing the number of genetic alterations, between the subclones is calculated using the Hamming distance[29] and phylogenetic trees are reconstructed using the maximum likelihood and parsimony methods. In addition, the algorithm provides the distribution and size of the clusters across the samples, resulting in an overview of the dynamics, spatial distribution, and dissemination of the tumor (Fig. 1g). The mean execution time of the algorithm is in the order of seconds.

**Validation on pediatric tumors confirms concordance to established evolutionary scenarios.** DEVOLUTION was applied to a previously reported dataset of 56 pediatric cancers and phylogenetic trees were generated based on copy number data for 22 neuroblastomas, 20 Wilms tumors, and 8 rhabdomyosarcomas comprising a total of 253 biopsies (Fig. 2, Supplementary Figs. 2–7, Supplementary Data 1 and 2)[17]. Event matrices but not

phylogenetic trees could be reconstructed from six patients (NB1, NB24, WT1, WT2, WT3, and WT5) in which all cells across the samples from the same patient had identical genomic profiles.

The phylogenetic trees of the remaining 50 tumors all represented plausible biological scenarios and often illustrated key events in tumor evolution, in line with previous studies[14]. For example, *MYCN*-amplification, known to be an early event in NB, was placed in the stem by DEVOLUTION in 7/7 tumors in which it was present. Among other early events of pathogenic importance in NB, whole chromosome 17 gain (+17) was placed in the stem in 8/9 cases where it was present. In the discrepant case (NB10) merely two biopsies were present of which +17 was 100% in one of them. 17q gain, was identified in 13 NBs of which it was placed in the stem in 8. In the 5 remaining tumors it was found in all biopsies but were not present in ≥90% in all of them, nonetheless indicating that the genetic alteration consistently presents early in the evolution of these tumors, in line with previous studies[14]. As for established early pathogenic drivers in RMS, 11p cnni was found in the stem in 5/6 tumors (in RMS2 it was found in the first branch after the stem). Both the *PAX3/FOXO1*-fusion in RMS7 and *PAX7/FOXO1*-fusion in RMS8 were confined to the stem. As for WT, 11p cnni was placed in the stem in 7/9 cases. The remaining cases were in one case a tumor only encompassing 3 genetic alterations of which the 11p alteration made up 60% of one of the biopsies. The other case was a complex case with 4 different types of cnni:s in 11p that were early in the branching of the phylogeny (Supplementary Figs. 2–7).

DEVOLUTION was also useful for tracing patterns of metastasis. In NB5 (Fig. 2a), where a primary tumor and a metastasis presented at the same time, the metastasis was demonstrated to originate from a population of cells having the genetic alterations of the stem as well as one group of cells encompassing a subclone also present in the primary tumor. This indicates polyclonal seeding. A more complex pattern of

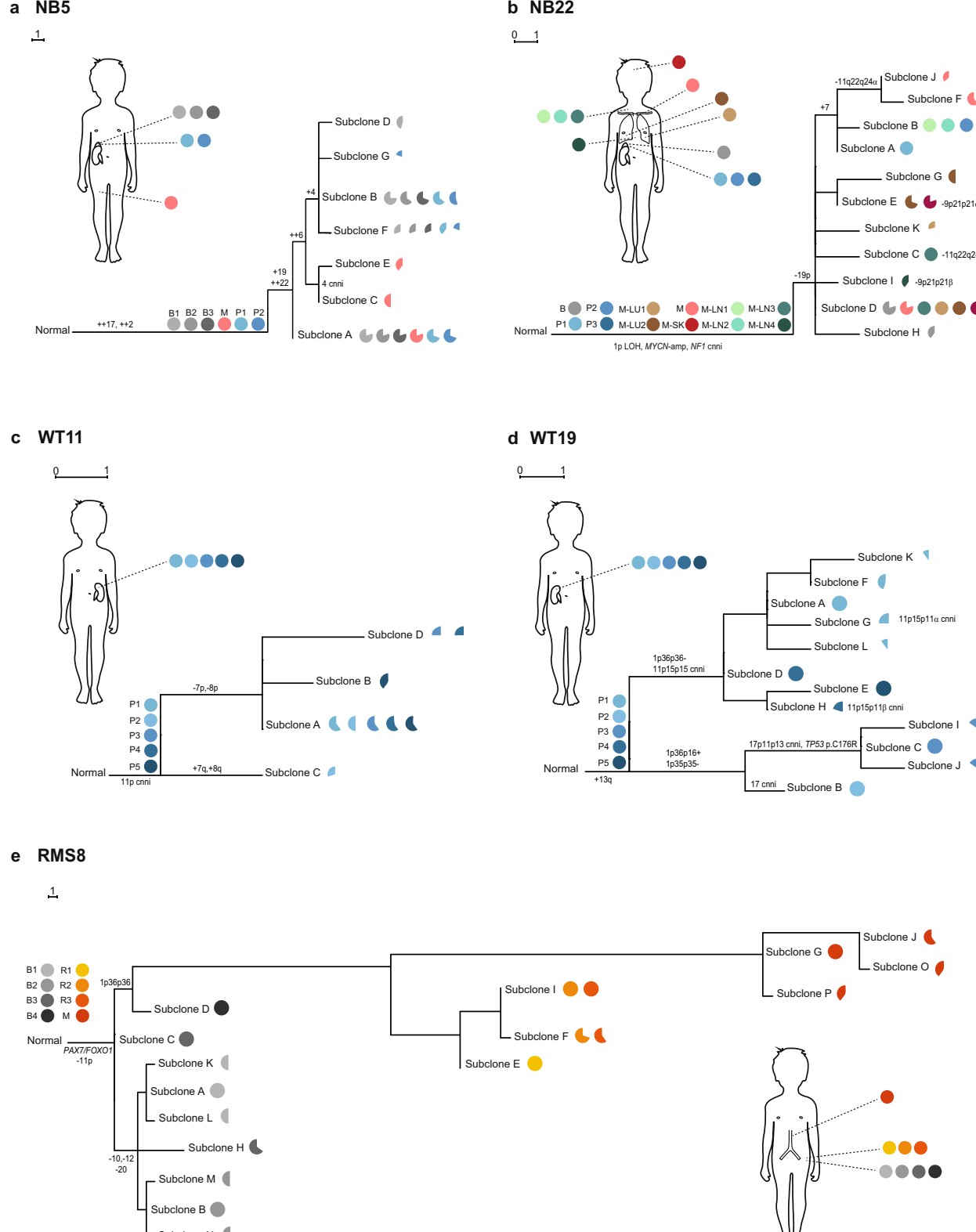

polyclonal seeding was observed in NB22 (Fig. 2b), a patient with progressive tumor growth across multiple metastatic locations. Here there was more subclonal variation among the lymph node metastases than among metastases to distant organs. The metastases to distant organs often presented as solitary branches, exemplified by the same subclone colonizing both the lung and skull. This might indicate that the threshold for tumor cells to

escape the primary tumor and colonize the lymph nodes is lower than that for colonization of distant organs, displayed as a wider variety of different subclones across lymph node locations compared to an extensive selection for a certain subclone in distant loci. RMS8, an alveolar rhabdomyosarcoma (Fig. 2e), displayed an intricate evolutionary pattern with many genetic alterations. Here the primary tumor's subclones form a cluster at

**Fig. 2 Phylogenetic trees of childhood cancers.** At the stem of the maximum parsimony trees illustrated here, the biopsies available from each patient are denoted. The genetic alterations belonging to the stem are present in all cells in all samples, indicated by filled pies. The endpoints represent cell populations harboring distinct genomic profiles (subclones), whose fractions per sample are visualized by pie charts. The scale bar indicates the distance corresponding to one genetic aberration. Gains and losses of chromosomes or segments of chromosomes that are characteristic of each tumor subtype are indicated by + and – signs. **a** In NB5 samples are available from the primary tumor before treatment (B1–B3), a synchronous metastasis (M), and the primary tumor post treatment (P1–P2). The metastasis must have originated from a subclone harboring the stem events only and another subclone with the copy number profile seen in subclone A, indicating polyclonal seeding. The metastasis also has a private copy number neutral imbalance (cnni) of chromosome 4. **b** NB22 also shows evidence of polyclonal seeding. Samples are from the primary tumor before treatment (B), the primary tumor post treatment (P1–P3), metastases to the lung (M-LU1-2), to the lymph nodes (M-LN1-4), the skull (M-SK), and from the area around the clavicle (M). The stem harbors a 1p cnni, *MYCN*-amplification and a *NF1* deletion. Greek letters denote different structural alterations targeting partly overlapping regions. **c** WT11 shows subclones present across multiple primary tumor areas (P1–P5). **d** WT19 displays a similar distribution of subclones as WT11 across the post-treatment biopsies P1–5. **e** In RMS8, subclones in the biopsies from a local relapse (R) and a distant metastasis (M) form their own branch harboring several additional genetic alterations compared to the primary tumor biopsies (B). The information used to produce the phylogenies can be found in Supplementary Data 1 and 2.

the root of the tree, while the cell populations from a metastasis and a local relapse share a branch having a vast amount of additional genetic alterations. As to evolutionary trajectories across the primary tumor space, WT11 and WT19 (Fig. 2c, d) both showed subclones that were distributed across several locations within the primary tumors, a phenomenon called subclonal coexistence which has previously been demonstrated to be common in Wilms tumors[17]. Hence, phylogenetic trees produced from copy number profiles by DEVOLUTION, can provide biological insights that might aid the understanding of how cancer develops and progresses in individual patients.

**Contradictions are rarely seen in the phylogenetic trees.** In 80% of all tumors analyzed, the maximum likelihood (ML) and parsimony (MP) methods resulted in identical phylogenetic trees (19/22 NB, 17/20 WT, 4/8 RMS) (Fig. 3a–d). When the ML and MP tree for the same case did differ from one another, the differences in the branching structure were minor (Supplementary Figs. 2–7). We identified the positions of the genetic copy number alterations in each tree to identify contradictions based on prior knowledge about how genetic aberrations occur in cancer cells. More specifically, we analyzed instances of PLC and BMC.

When the ML and MP trees differed from one another, this was always due to a PLC/BMC in the phylogeny, often with both contradictions together in the same tree. PLC and/or BMC were found in 14/50 ML-trees and 14/50 MP-trees (5 NB, 5 WT, 4 RMS), hence 28/100 trees in total. Of these, 3/5 NB, 3/5 WT, and 1/4 RMS trees contained only one single contradiction located among the leaves of the trees i.e., it did not have any significant impact on the tree structure. In addition, there did not seem to be any apparent difference between the frequency of the types of contradictions between ML and MP trees (Fig. 3e). Excluding the cases with PLC and BMC of whole chromosomes and chromosome arms, which are plausible events, only 8/50 ML-trees and 8/50 MP-trees (0 NB, 4 WT, and 4 RMS) exhibited contradictions in the tree structure. In these eight cases, merely a few genetic alterations were responsible for the PLC and/or BMC. They were caused by aberrations altering clone size compared to another event across samples or similar aberrations that still fell outside the breakpoint cutoff for similarity causing them to be considered separate events. These situations may be resolvable by critically reviewing the original data (Supplementary Fig. 8). Alternating clone sizes were particularly common in the RMS trees (Supplementary Fig. 9). The number of branches, total branch length, and stem length of the phylogenetic trees for the NB-, WT-, and RMS-tumors for MP and ML were compared using a two-sided Mann–Whitney U test. Statistically significant differences were seen between MP and ML for NB and WT compared to RMS for total branch length and stem length. Consequently,

the RMS tumors had a significantly higher total branch length and stem length than NB and WT (Fig. 3b), indicating a more complex genomic profile. They also had a mean number of genetic alterations per biopsy of 20, most of them present in >50% of cells in a single biopsy thus allowing just one single solution of the temporal evolution. These two aspects explain the residual difference between the MP- and ML-trees. To prompt review of original data when pertinent, the software will warn the user that there is a contradiction in the data set and the tree might therefore not be entirely biologically accurate.

**Evaluation using simulated data.** The reliability of the algorithm was further evaluated using simulated bulk sampling data. In the patient data set, the median number of unique subclonal alterations in total across all biopsies were 6 for NB, 5 for WT, and 13 for RMS. To accommodate this large variation, the simulation was conducted for three different mutation frequencies resulting in 15, 50, and 100 subclonal genetic alterations distributed across 40,000 virtual tumor cells (Fig. 4a). Virtual biopsies were sampled randomly from the set of cells while varying the number of biopsies from 1 to 10, generating a segment file along with a list of true unique subclones across biopsies for each mutational frequency. Hence, analysis could be performed using DEVOLUTION while having the true subclonal composition at hand.

As expected, when increasing the number of biopsies or the mutation frequency more genetic aberrations were identified. In addition, a higher number of subclones were correctly allocated (Fig. 4b). The performance for common clinical scenarios where merely one or two biopsies per patient is often available, are visible in Fig. 4b–c. A higher number of overall mutations will provide the software with more information, which is why the number of correctly allocated genetic alterations increase with mutation frequency for the same number of biopsies. Specifically, the number of positions in the temporal sequence at which genetic alterations can be allocated decreases with the number of genetic alterations. The number of unique subclones increases while the proportion of cells in each biopsy representing each subclone decreases. However, when more alterations are correctly allocated, in absolute numbers, the proportion of correctly allocated alterations will not increase. Throughout the different mutational frequencies 93 ± 5.5% of the genetic alterations were correctly allocated in the event matrix (Fig. 4c). Thus, when increasing the number of biopsies, the absolute number of correctly allocated genetic alterations increases, but the proportion of correctly allocated alterations does not change significantly. The reason for this is that sampling additional locations will also increase the chance of finding an area with a late genetic alteration that is hard to correctly place in the phylogeny because of its low spatial dissemination. We further dissected why not all

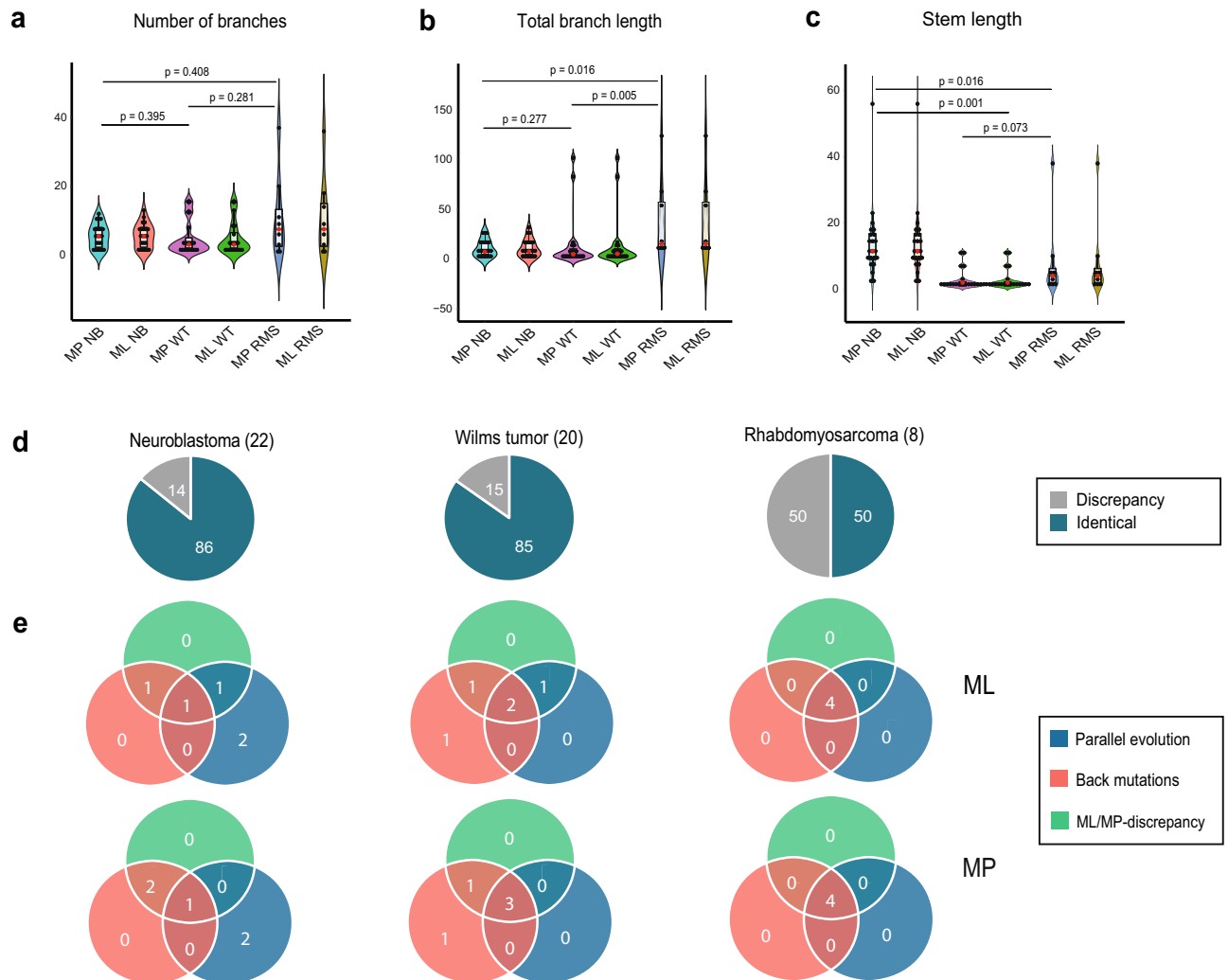

**Fig. 3 Structural properties of the generated phylogenetic trees.** Violin plots of **a** the number of branches, **b** total branch lengths, and **c** total stem lengths for neuroblastoma (NB), Wilms tumor (WT), and rhabdomyosarcoma (RMS) using either the maximum likelihood (ML) or parsimony (MP) method for phylogenetic reconstruction. Significance represented by P-values were calculated using the two-sided Mann–Whitney U-test. The box plots within the violin plots illustrates the interquartile range. The red dot is the median value. **d** In 14% of neuroblastomas, 15% of Wilms tumors, and 50% of the 8 rhabdomyosarcomas analyzed, the phylogenetic trees obtained using the ML and MP methods differed from one another. NB1, NB24, WT1, WT2, WT3, and WT5 are not included in this calculation since they did not display any private genetic alterations. Hence only event matrices could be generated but not phylogenetic trees. If including these cases, the proportions would be 12.8% for NB and 12.5% for WT. **e** Venn diagrams of how often discrepancies between ML/MP trees, back mutations, and parallel evolution occurred together in the same case/tree. Numbers indicate the number of tumors where a particular contradiction or combination of contradictions occurred. The information used to produce the plots can be found in Supplementary Data 1 and 2.

subclones were correctly allocated. It was hypothesized to occur due to low spatial dissemination, resulting in the presence of certain genetic alterations in a subset of biopsies. Excluding all genetic alterations found in only one single biopsy in fact resulted in the correct allocation of 99.5 ± 0.8% of the genetic alterations in this mixture of entities (Fig. 4d–e).

**Unifying sequencing and SNP-array data reveals additional evolutionary pathways**. DEVOLUTION with subsequent phylogenetic reconstruction was then applied to tumors in the pediatric cancer dataset for which SNP-array as well as sequencing data was available, including 8 NB (median 3 biopsies ranging from 2 to 7), 9 WT (median 3.5 biopsies ranging from 2 to 7), and 3 RMS (median 3 biopsies ranging from 2 to 8) (Fig. 5a–h, Supplementary Figs. 10–13). Phylogenetic trees were constructed based on sequencing data alone, SNP-array data alone, as well as with both

datasets in unison. Biopsies for which only information from one of the methods were available, were excluded from the analysis. NB18 (Fig. 5a–d) exemplifies how additional information concerning the evolutionary trajectory of the tumor is revealed when unifying the data types. The sequencing data revealed an *APO-BEC4*-mutation in the stem that is not identified using SNP-array. When analyzing the SNP-array data four different amplicons in the *MYCN* region in 2p24 as well as a 17q gain are identified, both of which are predictors of poor prognosis and aggressive disease in neuroblastoma that were not identified with sequencing data alone. Combining the data sets for subclonal deconvolution with DEVOLUTION conveys a unified picture providing a more detailed representation of the tumor's evolution. NB16 (Fig. 5e–h) is near diploid, disclosed by the diminutive SNP-array tree. The sequencing data tree, on the other hand, exhibits several mutations. Also, here a unifying picture captures additional aspects of the tumor evolution with a *MYCN*-amplification in the stem and a

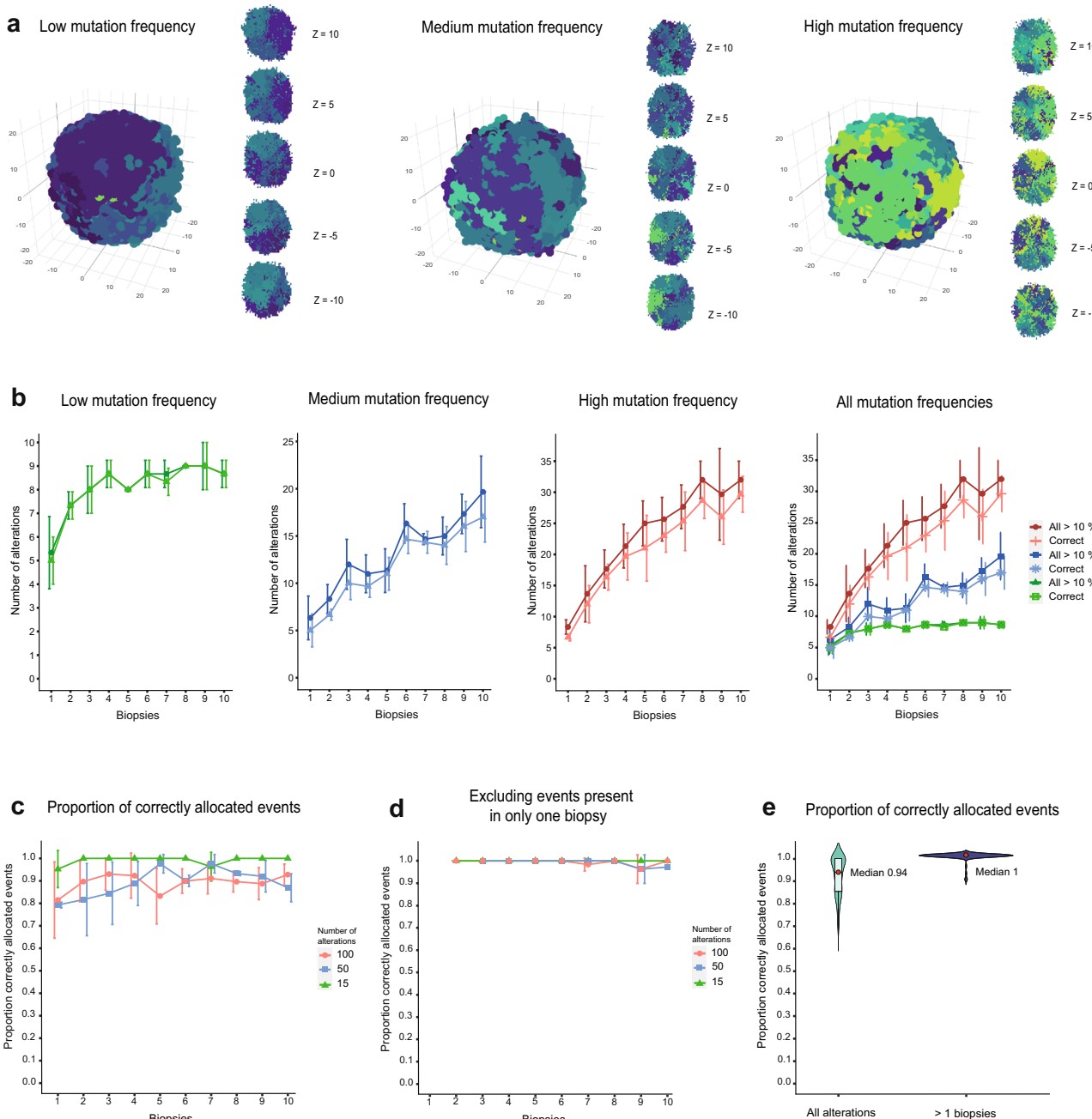

**Fig. 4 Properties of the simulated bulk genotyping data set along with the performance measure of DEVOLUTION. a** Visualization of three simulated tumors with increasing mutation frequency. To the right of each tumor are cross-sections at five positions (z). Each color represents a subclone harboring a unique genetic profile. **b** When increasing the number of biopsies, more genetic alterations and hence subclones, are identified. In addition, the algorithm is also able to identify more subclones correctly. Each point in the graph is the mean of three consecutive measurements and the error bars consequently the standard deviation. **c** The proportion of genetic alterations correctly allocated ± 1 SD when increasing the number of biopsies from 1 to 10 for three different mutation frequencies resulting in 10 (green), 50 (blue), and 100 (red) genetic alterations present in the virtual tumor. Each point is the mean of three iterations. **d** The proportion of correctly allocated genetic alterations when excluding genetic alterations that were only found in a single biopsy. **e** Violin plot showing the spread of the proportion of correctly allocated genetic alterations when including all alterations and when excluding the genetic alterations only found in one biopsy. The box plots within the violin plots illustrates the interquartile range. The red dot is the median value. The information used to produce the plots can be found in Supplementary Data 3 and 4.

subclonal *NRAS* mutation. Analyses of NB3 and NB7 (Supplementary Fig. 10) further emphasizes the importance of including copy number aberrations in the analyses of pediatric tumors. A two tailed Mann–Whitney U test was applied to compare the stem length, number of branches, and branch length for MP and ML based on data sets consisting of sequencing data, SNP-array, and sequencing data together with SNP-array data. Significant differences were found between the stem length for sequencing data MP/ML versus sequencing data together with SNP-array MP/ML as well as between the branch lengths for SNP-array MP/ML versus sequencing data together with SNP-array MP/ML for neuroblastoma, with trends towards similar differences for the other tumor types (Supplementary Fig. 13). Consequently, unifying the genetic information obtained from sequencing data and

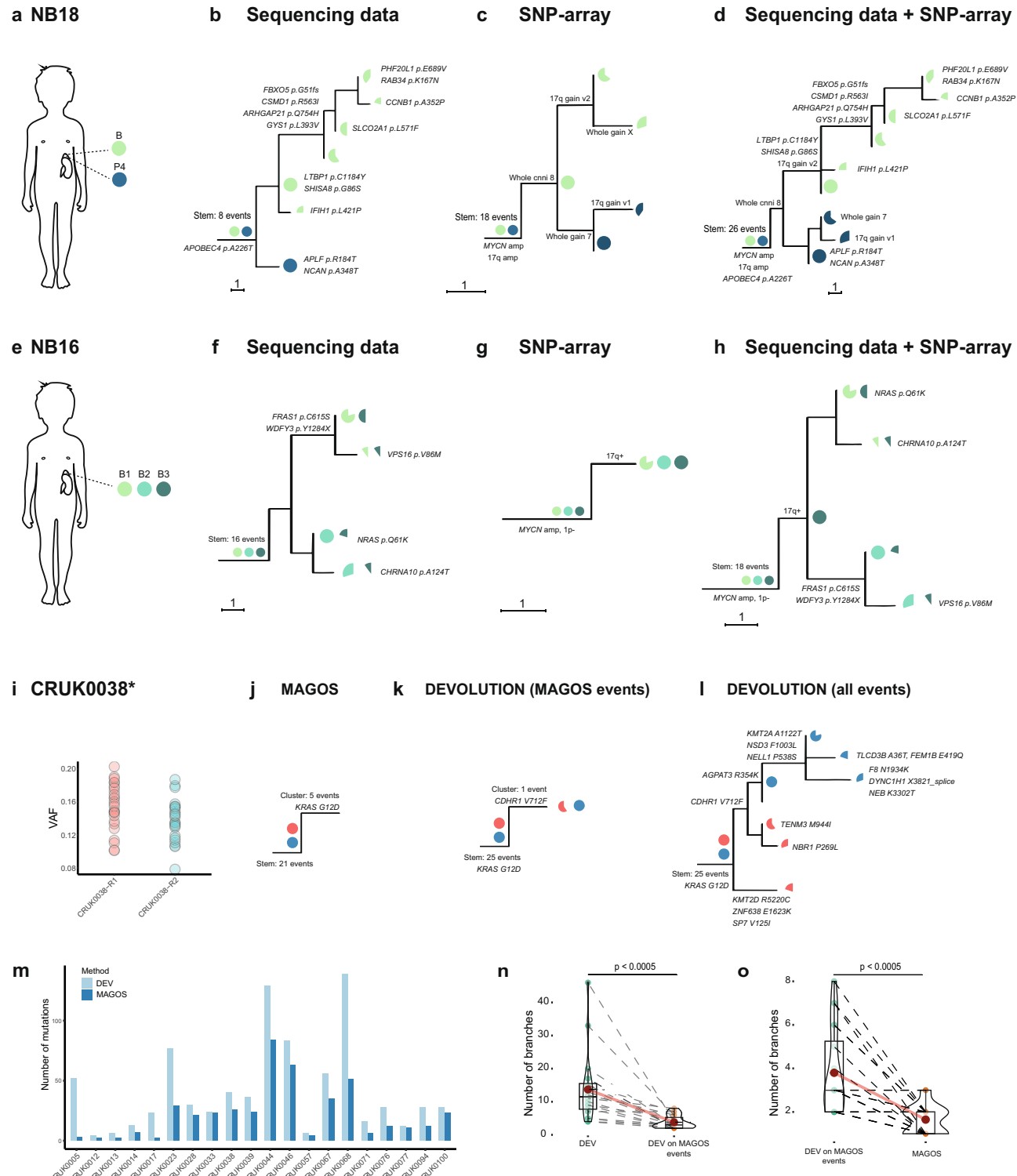

SNP-array further elucidates details in evolutionary trajectories. Combining information from multiple data types also allows for the construction of phylogenetic trees even if limited data is obtained with one method on its own.

**Using an external dataset for benchmarking**. The performance of DEVOLUTION was finally evaluated on a typical solid adult tumor, in comparison to MAGOS, which has been shown to outperform existing clustering algorithms such as PyClone and

SciClone[30]. The publicly available TRACERx data set for non-small-cell lung cancer (NSCLC) provides WES profiles for each tumor, some multi-regionally, along with copy number profiles for a subset of tumors[31]. WES data was extracted from 20 tumors for which multiple biopsies were available along with copy number profiles for each biopsy. Only cases where a sufficient number of mutations passed quality control for the mutations were included, defined as a read depth ≥10 and total coverage ≥200 reads for each mutation across all samples. Further, only mutations present in diploid segments in all samples in which it

**Fig. 5 Multimodal phylogenetic trees and application on adult cancers. a** NB18 including samples from the primary tumor at diagnosis (B) and after treatment (P4). **b** Phylogenetic reconstruction using sequencing data alone reveals an *APOBEC4* mutation in the stem while **c** SNP-array identifies a *MYCN*-amplification and 17q gain. **d** Combining the data sets gives a refined description of this tumor's evolution. **e** NB16 including three biopsies from the tumor before treatment (B1-3). **g** The tumor is near diploid as seen in the phylogenetic tree comprising the SNP-array data, **f** while having a larger amount of somatic point mutations. **h** Also, here combining the data sets gives a more refined phylogenetic description of the tumor. Panel **i–l** illustrates a phylogenetic analysis of the NSCLC CRUK0038. **i** A scatterplot illustrating the VAF of the mutations across the biopsies R1 and R2 for the NSCLC CRUK0038. The corresponding phylogenetic trees are based on **j** MAGOS alone, **k** DEVOLUTION on the events included in the MAGOS analysis (events present in all samples) as well as **l** the tree obtained with DEVOLUTION including events present in the individual biopsies. **m** The number of mutations includable in analysis using DEVOLUTION compared to MAGOS. **n** The number of phylogenetic branches produced using DEVOLUTION on the full data compared to DEVOLUTION on the events includible in the MAGOS analysis. Dotted lines connect data points from the same tumor. *P*-values were computed using a two-sided Mann–Whitney U-test. **o** The number of branches using DEVOLUTION on the events includible in the MAGOS analysis compared to the number of branches obtained in the phylogenetic tree based on nesting of the MAGOS clusters. Dotted lines connect data points from the same tumor. *P*-values were computed using a two-sided Mann–Whitney U-test. The box plots within the violin plots illustrates the interquartile range and the red dot is the median value. The information used to produce the phylogenies can be found in Supplementary Data 5 and 6.

existed were included in the analyses. The 20 NSCLC tumors had a median of 3.5 (median absolute deviation (MAD) 0.5) biopsies and 98.25 CNVs per biopsy (MAD 45.375). Since MAGOS can only analyze somatic point mutations present in all biopsies analyzed, a truncated data set was constructed fulfilling this requirement for MAGOS. DEVOLUTION can, on the other hand, include mutations present in a subset of biopsies as well.

Compared to MAGOS, DEVOLUTION revealed in higher detail the spatial heterogeneity of the analyzed NSCLCs. As an example, for CRUK0038 (Fig. 5i–l) in total 26 unique mutations were present in both biopsies (R1 and R2). MAGOS clustering (Fig. 5j) resulted in two clusters of similar size (variant allele frequency (VAF) 0.15 and 0.1, respectively, in R1 and 0.13 and 0.11 in R2). Here, MAGOS assigned the *KRAS* mutation as being subclonal despite high VAFs (0.13 in R1 and 0.18 in R2) in both samples while DEVOLUTION used on the same set of mutations inferred it as clonal (Fig. 5k). When DEVOLUTION was employed for analysis also of mutations present in merely a subset of the biopsies, this revealed a more elaborate phylogenetic architecture than was possible to obtain with MAGOS (Fig. 5l). The number of mutations possible to include were consistently higher for DEVOLUTION compared to MAGOS across the NSCLC (Fig. 5m, Supplementary Figs. 14–17). The total number of branches in the phylogenetic trees were also significantly ($p < 0.0001$) higher using DEVOLUTION compared to MAGOS (Fig. 5n–o, Supplementary Figs. 14–17, Supplementary Data 4) using a two-sided Mann–Whitney U-test. Also, MAGOS not including mutations that were regionally localized, occasionally resulted in mutations specific for metastatic lesions along with information on the primary tumor origin being disregarded (Supplementary Fig. 15 CRUK0013), which may hamper the analyses of timing of mutations during the metastatic process.

**Alternative solutions for the phylogenetic tree structure.** There may be situations where multiple phylogenetic trees are able to explain the observed data. An algorithm was therefore constructed to assess alternative solutions of the evolutionary trajectory of the genetic alterations. In the case where there is more than one solution, the user is asked if an alternative solution should be shown in addition to the suggested solution by DEVOLUTION. In that case, the clusters of genetic alterations unique for those subclones that have multiple solutions are first removed from the tree structure and then randomly reshuffled to produce a new phylogenetic tree, that does not violate any of the rules in any of the samples or rules provided by the user (Supplementary Figs. 18 and 25, Supplementary Methods). The user is also provided with a matrix illustrating which subclones in the tree are reliable and which are uncertain due to multiple possible evolutionary trajectories.

## Discussion

A single biopsy from a tumor can contain multiple distinct subclones and their prevalence may vary across biopsied areas. Not addressing this fact when studying cancer cell evolution can be deleterious and result in incorrect phylogenies[15], stressing the need for multispatial and temporal sampling to unfold the genomic landscape. DEVOLUTION thoroughly assesses the problem by combining information obtained across multiple biopsied regions throughout the entire subclonal deconvolution, effectively deconvolving subclones transversing clonal territories, with the potential to concomitantly include both point mutations and copy number alteration data. In contrast to other methods DEVOLUTION allows phylogenetic trees to be constructed using copy number information alone and integrating information from multiple biopsied areas throughout the inference procedure. The algorithm can combine data from SNP-array, TDS, WES, and WGS, provided the MCF for each genetic alteration is known, which can be computed based on the log2 ratio for copy number alterations or variant allele frequencies (VAF) for point mutations as described extensively elsewhere[17,21,22]. Since aneuploidy is a common feature across many adult carcinomas and a majority of childhood cancers, the integration of copy number aberrations in the phylogeny holds the potential to increase the understanding of the evolution of these diseases. If needed, there is also a possibility for user curation, for example if it is known that certain genetic alterations cannot co-exist.

Evaluating the software using high-throughput SNP-array data from 253 multitemporal and spatial regions from 56 pediatric tumors produced phylogenetic trees that were in concordance with prior knowledge of how chromosomal aberrations occur in cancer cells[14]. Surprisingly, generating phylogenetic trees using ML and MP predominantly yielded identical tree structures. When thoroughly examining these trees, contradictions such as PLC and BMC were identified in the cases where the ML/MP trees differed, which were found to always be due to disagreements in the original data set. The user is therefore encouraged to reevaluate this genetic alteration in the input segment file, since it may be particularly subjected to noise or keep the tree if the phylogenetic situation is considered biologically plausible. The extensive evaluation of the two methods did not indicate that the choice of mathematical method favors a certain type of error in the phylogenetic tree. We found, however, that the RMS tumors had a significantly higher branch length compared to NB and WT, suggesting a more complex genomic profile—compromising the possibility of robust clonal deconvolution.

Clustering genetic alterations using DBSCANs was sufficient for analyzing the pediatric tumors incorporated in this study and the simulated data sets. The size of $\epsilon$ can be changed by the user to increase the flexibility in the clustering, thus optimizing it

further for the data set at hand to account for noise. The purpose of the ad hoc clustering mainly is to reduce the computational complexity and not to find clones and is not to be confused with the clustering used in dedicated clustering algorithms for clonal determination. The choice of clustering method can also easily be changed by the user.

While DEVOLUTION enables further analysis of heterogenous multiregional and temporal tumorigenic data there are still venues for improvements. Firstly, using the MCF as an input for the algorithm stresses the need of a robust pre-analysis of the data set. The inferred subclones identified could be affected by the choice of method for computation of the MCF-values. Methods to compute these has been discussed extensively elsewhere[17,31,32] and novel methods are being developed[33,34]. The simulations performed in this study show that if the MCF-values are correctly inferred, the algorithm provides a robust nesting and prediction of the subclones in the samples. Secondly, for cases where solely stem events are identified, creating a phylogeny will not provide the user with additional information since the phylogeny will just be a horizontal line. Alternatively, for cases where a vast number of genetic alterations are found, such as in the RMS, the noisiness of the MCF computation becomes more apparent and events that cross over one another in sizes across samples becomes more common, which will aggravate the performance of DEVOLU-TION. Increasing the number of mutations in your data set will require a stronger preclustering to reduce the occurrence of PLC and BMC. Events that cross each other in size across samples are problematic since there is no way to elucidate in which order they appeared in the evolutionary history of the tumor. The user is in these cases advised to revisit the data to make sure it is biologically feasible, since it implies parallel evolution or back mutations of the events in question.

Much progress has been made in the field of clonal deconvolution. However, many methods are limited to integrating information from a single biopsy, such as TITAN and THetA in addition to only accepting sequence data[35,36]. PyClone and Sci-Clone are, however, employable on multiple biopsies, but assume that all detected CNAs are clonal i.e., present in all cells in each biopsy, do not infer the evolutionary relationship between the identified clusters, and require sequence data to operate[21,22]. In addition, SciClone focuses exclusively on sequence variants in copy number neutral and loss of heterozygosity (LOH) free portions of the genome. Both PyClone and SciClone mainly operate as clustering algorithms and do not infer the order of genetic alterations. Their output could, in fact, be used as an input to DEVOLUTION. Hence, DEVOLUTION and PyClone/Sci-Clone fulfill different purposes and are not meant to exclude one other. PhyloWGS on the other hand can use both sequence variants and CNAs to infer a phylogeny and is employable on multiple samples. However, the algorithm does not integrate information between samples during the inference procedure, representing a loss of valuable information, and is limited to WGS data[20]. Also, other methods such as Clomial, LiCHeE, and SCHISM are specifically designed for SNVs and cannot solely include CNAs to infer a phylogeny[37–39]. REVOLVER also requires sequencing data, cannot use CNA-data alone, and is specifically designed to integrate phylogenies from large cohorts of patients to infer common trajectories of repeated evolution[40]. SPRUCE requires input of sequencing data to infer phylogenetic trees[41].

In contrast to these sequence-oriented applications, MEDICC is specifically designed for CNA evolution but does not allow inclusion of point mutations; the inferred phylogenies are solely sample trees and can only handle copy numbers up to 4[42]. TuMult is also limited to sample trees and cannot integrate point mutations[43]. HATCHet allows for copy number variations but

requires whole genome sequencing data. It looks at copy number profiles jointly across multiple tumor samples from the same patient. However, it does not reconstruct a phylogeny of the identified genetic alterations. The output of HATCHet is number states and the clone proportion indicating the fraction of cells in a sample having that particular alteration. Hence the output of the HATCHET algorithm, which provides an intriguing improvement of current methods, can be used as the input for DEVOLUTION[44]. Taken together, most methods available focus exclusively on sequencing data and there is currently no available tool that can reconstruct a phylogenetic tree based on multiregional SNP-array data alone, which is a commonly used genotyping method in the clinic. Additionally, many methods focused on sequence variation only produce clusters of genetic alterations and their MCF. Event matrices are subsequently often constructed manually from these MCF estimates, posing a risk for unintentional subjective bias in the deconvolution process, especially when integrating information from multiple biopsies. These methodological gaps are filled by DEVOLUTION, which provides an objective method to infer phylogenies based on a priori MCF estimations based on preset rules that are employed equally across all patient data, hence providing a standardized framework for inferring phylogenies from bulk genotyping data, thus allowing tree-to-tree comparison without the risk of bias from subjective curation.

In summary, we have seen how DEVOLUTION can be used to analyze the intratumoral heterogeneity as well as the intertumoral heterogeneity between the primary tumor and metastasis through evaluation on a dataset of pediatric tumors harboring extensive aneuploidy. By analyzing a cancer's phylogenetic tree, an overview of its heterogeneity and temporal order of genetic alterations can be assessed, which can be used to follow the tumor's evolutionary response to treatment[45,46] and may aid the identification of subclones posing a risk of metastasizing, relapsing, or being resistant to therapy. It may also make it possible to identify genetic alterations that seem to appear early in the tumor's development, posing attractive targets for therapy since they are present in a large proportion or all cells in the tumor.

## Methods

**Ethics statement—Pediatric tumor data set.** The present study complied with all relevant laws and ethical guidelines regarding human research participants. Analysis of human tumors was approved by the Regional Ethics Board of Southern Sweden (permit nos. 119-2003 and 289-2011). Patients were selected after written informed consent from themselves, their parents, or legal guardians from a cohort of pediatric cancer patients treated at the Skåne or Karolinska University Hospitals in Sweden[17].

**The software**. The major structure of the software (Supplementary Methods, Supplementary Figs. 19–29) can be divided into five steps

1. Preprocessing of the data.
2. Clustering of genetic alterations based on information from multiregional sampling from the same patient.
3. Subclonal deconvolution based on information from multiregional sampling from the same patient.
4. Construction of an event matrix.
5. Usage of a mathematical model to reconstruct the phylogenetic trees, in this case maximum likelihood and maximum parsimony.

**Preprocessing of the data**. The input data consist of an $u \times v$ dimensional matrix containing information about the $u$ detected genetic alterations present in each biopsy. The matrix should also specify the genetic location of each alteration, its type (gain, loss, cnni etc.) as well as the proportion of cells harboring the alteration in that particular biopsy (the mutated clone fraction, MCF), represented by the $v$ columns (Supplementary Table 2). There are multiple dedicated tools that can infer MCFs from sequencing data such as the clustering algorithms PyClone, SciClone, and MAGOS, and recently also DeCiFer[33] and for structural variants SVclone[32] which could be used as an input for DEVOLUTION. How the MCF can be

computed from log2 ratios for copy number aberrations as well as VAFs is described extensively elsewhere[17].

If allelic copy number alterations are considered, the user is advised to choose a cutoff for the detected genetic alterations in the segment file to be considered separate events, reflecting the measurement uncertainty regarding the start and end positions of the genetic alterations. The default cutoff is 1 Mbp. The user can also choose which data types to include in the analysis. In this way e.g., SNP array and sequencing data can be analyzed separately for comparison or in unison without having to separate the matrix manually.

The algorithm scans the MCFs for missing values, indicating that the MCF has not been able to be determined. If the event is considered to belong to the stem, based on biological knowledge or additional data, the missing value is replaced by 100%. Amplicon accumulation is an example of such a case when it is not possible to determine the fraction of cells harboring it since the number is hypervariable. A stem event is defined as the presence of the alteration in ≥90% of the cells in all samples. The alterations containing missing values for MCF are removed entirely if part of a subclone to not overestimate genetic variation within the tumor.

The clustering algorithm was constructed to localize all unique genetic alterations throughout the tumor samples. The program loops through the rows of the data file representing the genetic aberrations. For each row it compares the genetic alterations and their position on the chromosome to all the other rows, representing other detected genetic aberrations throughout the samples. If the events' start or end positions differ by a certain cutoff, set by the user based on the measurement uncertainty of the data set, and/or they are different aberration types, they are considered as two separate events, else they are considered as the same event (Supplementary Fig. 1). Thus, all conditions stated below must be met for the algorithm to consider two alterations detected in the same patient to be the same.

1. Alteration 1 and 2 are localized on the same chromosome.
2. Alteration 1 and 2 harbor the same type of alteration.
3. Neither alteration 1 nor 2 should belong to the stem.

Alterations belonging to the stem are always considered as separate events.

$$X_1 = \|A_{1_{start}} - A_{2_{start}}\| \le co_{ev}$$
$$X_2 = \|A_{1_{end}} - A_{2_{end}}\| \le co_{ev}$$

In the present study, considering allelic copy number aberrations, the cutoff ($co_{ev}$) for measurement uncertainty in the start and end position of the events was set to 1 Mbp, also constituting the default for DEVOLUTION. Since the chromosome sizes range from 48 to 250 Mbp, this cutoff constitutes a start and end point deviation of 0.4–2% of the chromosome length.

**Clustering of genetic alterations incorporating information from multiregional and temporal sampling from the same patient.** In our model a tumor is proposed to consist of multiple subpopulations of cells that harbor different sets of genetic alterations. Each individual alteration is part of a mutation space $m_i \in \{m_1, m_2 \dots m_\theta\}$ comprising all mutations present in the tumor where $i, \theta \in \mathbb{N}^+$ and $\theta$ is the total number of mutations. The mutational profile obtained from the biopsies thus represent a subset of the total mutation space and is the information at hand to describe the evolutionary trajectory of the tumor. For this purpose, all detected mutations are combined with their respective MCF-values into a matrix representing their distribution across samples (Supplementary Fig. 30). For a particular tumor, this results in a matrix $T_{MxB}$ with the dimensions M× B, where M is the total number of unique genetic alterations and B is the total number of biopsies available. Hence $m_\delta$ indicates a certain genetic alteration $\delta$, and $b_\omega$ represents a biopsy $\omega$. The value $t_{\delta\omega}$ consequently corresponds to the MCF for an alteration, $m_\delta$, in a sample, $b_\omega$ where $t_{\delta\omega} \in [0,100]$ i.e., it is bound between 0 and 100%. This can be written as

$$T_{MxB} = \begin{bmatrix} t_{1,1} & t_{1,2} & \cdots & t_{1,B} \\ t_{2,1} & t_{2,2} & \cdots & t_{2,B} \\ \vdots & \vdots & \ddots & \vdots \\ t_{M,1} & t_{M,2} & \cdots & t_{M,B} \end{bmatrix} \quad (1)$$

where $t_{\delta\omega} \in [0, 100], \delta \in \{1, \dots, M\}, \omega \in \{1, \dots, B\}$ and $\delta, \omega \in \mathbb{N}^+$

In order to generate phylogenetic trees illustrating the relationship between the subclones present in the tumor, which aberrations reside in the same cells as well as which subpopulations of cells the tumor consist of must be determined. To solve this, the idea is that a true subclone of cells should form a cluster of unique genetic alterations that persist is adopted. They should remain grouped irrespective of inclusion of new data from an additional region of the primary tumor or metastasis. Alterations that seem to follow each other are more likely to be in the same cells. The first step is thus to yield a clustering to identify groups of genetic alterations, uniquely identifying a certain subclone. The subsequent step is to determine the temporal order of the alterations in question, since each subclone will represent a linear combination of the clusters identified. Note that for DEVOLUTION the clustering is only used to reduce the computational complexity for the upcoming subclonal deconvolution algorithm. Mostly alterations are clustered that show similar MCF in all available biopsies. This is not to be confused with the more intricate clustering methods used in for example SciClone.

Density-based clustering techniques such as DBSCAN[27] are superior at unsupervised clustering of non-uniform clusters. Furthermore, the number of clusters does not have to be specified beforehand, which you have to do with many other established clustering algorithms. In addition, it does only have two hyperparameters named *minPts*, which is the minimal number of points that is allowed in a cluster, and $\in$ representing the radius in which points i.e., the genetic alterations' position in the B-dimensional space, are included, where B is the total number of biopsies. The default value of $\in$ is 0.5 for DEVOLUTION if no other value is provided by the user through the input command to the function. Running the algorithm also yields a k-distance-graph which illustrates the distance to the minPts-1 = k nearest neighbor along with a designation of where $\in$ = 0.5 is located. The optimal value is where this plot shows an elbow. If the default value differs from this, for this particular data set, the user can provide the input function with a new epsilon, overriding the default parameter, for optimal clustering. The clustering method is well-confined within the algorithm and is therefore easy to replace with another method if the user finds so suiting[27].

The algorithm provides a matrix containing all clusters of genetic alterations. Let $C_{KxN}$ be the matrix representing the clusters of genetic alterations. It has the dimensions K x N where K is the number of genetic alterations in the cluster and N is the cluster number. All matrix positions $c_{kn} \neq 0$ are unique i.e., the same genetic alteration cannot belong to multiple clusters.

$$C_{KxN} = \begin{bmatrix} c_{1,1} & c_{1,2} & \cdots & c_{1,N} \\ c_{2,1} & c_{2,2} & \cdots & c_{2,N} \\ \vdots & \vdots & \ddots & \vdots \\ c_{K,1} & c_{K,2} & \cdots & c_{KN} \end{bmatrix} \quad (2)$$

where $c_{kn} \in m_\delta \wedge c_{kn} \neq c_{ed}, (\forall k, e \in \{1, \dots, K\} \& n, d \in \{1, \dots, N\} \wedge c_{kn} \neq 0)$

A matrix representing the clusters present in each biopsy and their size determined by the mean of the aberrations in the cluster is also constructed.

$$Z_{CxB} = \begin{bmatrix} z_{1,1} & z_{1,2} & \cdots & z_{1,B} \\ z_{2,1} & z_{2,2} & \cdots & z_{2,B} \\ \vdots & \vdots & \ddots & \vdots \\ z_{C,1} & z_{C,2} & \cdots & z_{CB} \end{bmatrix} \text{ where } z_{cb} \in [0, 100] \quad (3)$$

where c is a specific cluster of aberrations, b the biopsy and $z_{cb}$ the size of the cluster c in sample b.

**Subclonal deconvolution based on information from multiple samples from the same patient.** The space of a single biopsy is 100% and the space of all biopsies can thus be represented by a matrix where p is the partitioning of the available space in the biopsy and $s_{pb}$ is the space available in a specific partitioning p in biopsy b. Initially $s_{1,b} = 100 \wedge s_{p \neq 1,b} = 0$.

$$S_{PxB} = \begin{bmatrix} s_{1,1} & s_{1,2} & \cdots & s_{1,B} \\ s_{2,1} & s_{2,2} & \cdots & s_{2,B} \\ \vdots & \vdots & \ddots & \vdots \\ s_{P,1} & s_{P,2} & \cdots & s_{PB} \end{bmatrix} \text{ where } s_{pb} \in [0, 100] \text{ and } \sum_{p=1}^{P} s_{p,b} = 100 \wedge b \in \{1, \dots, B\} \in \mathbb{N}^+$$

(4)

The clusters of aberrations in each biopsy, as supplied by $Z_{CxB}$, are allocated to the space in decreasing order, altering the magnitude of the spaces in $S_{PxB}$ based on the MCF of the clusters allocated to it.

The allocation iteration algorithm is initially conducted considering each sample individually, resulting in a matrix encompassing all possible allocations of each cluster in every biopsy. Subsequently, all possible allocations throughout the samples are addressed to minimize the occurrence of parallel evolution. The algorithm hence tries to produce one uniform solution of the temporal order of events that does not contradict any information provided in the biopsies. The solution should be in concordance to every biopsy provided. If not possible, parallel evolution or back mutations will occur in the final phylogenetic tree and the user is advised to reconsider the original data set, since it may be biologically unlikely. It may be possible to allocate a cluster to multiple positions without producing contradicting temporal orders in any of the samples, for which the largest available space assumption is employed to make an objective decision, based on the presumption that the mutational frequency is equal in all cells within the biopsy no matter how many mutations they have. The cluster will consequently be placed as a descendant to the cluster constituting the largest proportion of the biopsy, in the absence of further biological information steering it elsewhere. Clusters presenting with only one possible allocation in a biopsy provides especially valuable information concerning the temporal order of events, e.g., having a group of alterations that are all present in all cells in a biopsy clearly indicates that there exists a group of cells in the tumor harboring all of these alterations, aiding the temporal allocation in other samples where these alterations may present themselves as subclonal. DEVOLUTION does allow some overlap in cellular frequency in the allocation algorithm taking into consideration the measurement uncertainty of MCF. This iterative computation results in the subclones present in

the biopsies along with an estimation of their size and distribution across the samples.

**Incorporating user-controlled rules for avoiding imposition of illicit biological trajectories**. Some genetic aberrations present in the data set might be known to never occur in the same cell for some well-known biological reason. Such constraints should optimally be supplied to the algorithm to ensure biologically plausible solutions. The user can therefore provide the DEVOLUTION algorithm with a matrix indicating which genetic aberrations in the data set that cannot be placed after one another. The first column represents a mother genetic alteration that the daughter alteration specified in the second column, cannot have (Supplementary Fig. 31). The subclonal deconvolution algorithm extracts a list for each genetic alteration containing information about in how many of the samples it can be allocated after a certain cluster. There might be multiple possible solutions, equally prevalent. In this instance the matrix containing information about illicit biological orders can aid the program in taking a decision regarding which of these allocations are less likely, subsequently discarding them. These rules will thus only be employed if the data set allows the genetic alterations to be placed in any other way. If the only possible way for the events to be allocated is to place them as descendants, the user will be advised to revise the original data set. No such rules were integrated in the analysis of the 56 pediatric tumors in the present study.

**Construction of an event matrix**. Based on the estimated subclonal composition, an event matrix $E = [\hat{a}_1, \hat{a}_2 \ldots \hat{a}_k]$ was constructed illustrating the distribution of genetic alterations across the identified subclones. Each $\hat{a}_i$ is binary vector belonging to subclone $i$. Each row represents a genetic alteration indicated with a 1 if present or 0 if absent in the subclone. The event matrix is used as the foundation for phylogenetic tree generation, illustrating the relationship between the subclones within the tumor.

**Reconstruction of phylogenetic trees**. In order to generate the phylogenetic trees, the genetic distances between all the subclones must be computed. Here, the Hamming distance between the subclones in the event matrix was used to assess a distance matrix displaying the genetic distance between each of the subclones. It computes the distance between two vectors by adding all positions in which they differ from one another, in this case the number of genetic alterations or positions in the event matrix the entities differ from one another, resulting in branch length in units of number of aberrations. No bias in the estimation of branch lengths were seen for ML in this study.

The user can choose in which entity the phylogenetic tree is to be rooted in. The default is a normal cell containing no genetic alterations. The event matrix is then transformed into phyDat format using the function *phydatevent*. This is the data class needed for phylogenetic analysis using the R package phangorn[47].

In the next step the maximum likelihood and maximum parsimony algorithms were used in order to reconstruct phylogenetic trees based on the event matrices. Both of which are well established methods.

The maximum parsimony method assumes that genetic alterations are rare and reconstructs the tree that requires the smallest number of evolutionary steps to explain the data, thus minimizing the occurrence of homoplasy, but does not forbid it. There are although multiple examples of homoplasy across species, such as the extensive convergent and regressive evolution of the traits of the cave fish[48]. In the context of point mutations and intrachromosomal copy number aberrations this is rare, but evolutionary pressures could possibly result in selection of similar phenotypical manifestations while maintaining an unchanged genotype. The maximum likelihood method strives to find the tree that maximizes the likelihood of obtaining our data set. The maximum likelihood trees were reconstructed using the pml algorithm in the package phangorn[47]. First a Hamming distance matrix was calculated from the event matrix which was used to obtain an initial tree given by the neighbor joining method. The initial tree as well as the initial event matrix was used as input variables in the pml algorithm. This function returns an object containing the tree parameters, the data as well as the likelihood for that phylogenetic tree. In order to optimize the tree parameters further, the function optim.pml was used in combination with the Jukes Cantor model, assuming equal transition rates and equilibrium frequencies for all states, and optEdge = TRUE. The tree was subsequently rooted in a constructed cell having all the events shared between all subclones. Since the model used is time reversible the choice of the root does not influence the computed likelihood[49]. The tree was visualized using the ggtree package in R[50].

The maximum parsimony trees were constructed using the parsimony ratchet algorithm (pratchet) in the R package phangorn with the Fitch algorithm[51]. Using the acctran algorithm the branch lengths and the ancestral character probability distributions were obtained. The trees were rooted in a cell containing no alterations.

**Performance testing using simulated bulk sampling data**. In order to further assess the reliability of the algorithm, it was evaluated using simulated bulk genotyping data using a basic 3D-lattice based, stochastic model of tumor growth. The simulation is initiated imagining a cell having one single genetic aberration, representing a stem event, which will be conveyed to all cells comprising the virtual

tumor. In each time unit one cell can proliferate. When a cell has been chosen for proliferation a certain inherent mutation frequency determines whether the cell will mutate or not. If not, two cells identical to the mother cell are obtained, otherwise a stochastic genetic copy number aberration algorithm is used to randomly select a genetic alteration (copy number alteration), conjointly considering the chromosome boundaries and sizes. Making use of a random number generator, a chromosome is randomly selected, then a start and end position and finally the type of event. The result is one cell identical to the mother and one cell harboring one additional genetic aberration. The spatial orientation of the cells is also considered where each cell is assumed to have 26 neighbors and two cells cannot occupy the same position. The position of the second daughter cell is randomly selected among the available neighbor positions. The simulation was conducted generating 40,000 cells giving a simple 3D lattice structure illustrating the spatial intratumoral heterogeneity for three different mutation frequencies (Fig. 4a). Note that only the fractions of cells harboring a certain genetic alteration is of importance in this model and not any absolute numbers. The goal is not to correctly simulate tumor growth but to obtain a random mixture of entities to be demixed using DEVOLUTION. The mutation frequency chosen, 10/40,000, 50/40,000, and 100/40,000, is not biologically accurate but chosen such that a certain number of mutations will appear during the simulation, resulting in segment files resembling the MCF-distributions seen in the patient cases.

Virtual biopsies were drawn from the set of simulated entities while varying the number of biopsies from 1 to 10. The biopsies were drawn randomly from different parts of the simulated entities. Values for $x$, $y$, and the $z$ coordinates were randomly chosen while fulfilling

$$x \in [x_{min}, x_{max}], y \in [y_{min}, y_{max}], z \in [z_{min}, z_{max}] \qquad (5)$$

$$\sqrt{x^2 + y^2 + z^2} \in \left[ \frac{r_{mean}}{2}, r_{mean} \right] \in R \qquad (6)$$

where

$$r_{mean} = \frac{x_{\max} + x_{\min} + y_{\max} + y_{\min} + z_{\max} + z_{\min}}{6} \qquad (7)$$

Based on the position chosen, the entities within a radius of 2 units were extracted and the fraction of cells harboring each of the alterations found was calculated. The data from each cell can be used to create artificial bulk sampling with MCFs as well as single cell data. Hence, the segment files can be analyzed while having the true subclones at hand for comparison. DEVOLUTION was evaluated using 1–10 biopsies for the three tumors.

**Computation of the mutated clone fraction (MCF)**. For each tumor, clustering was made with MAGOS. Since the clusters obtained are not subclones but merely collections of mutations with similar VAF, manual nesting was subsequently performed. For analyses with DEVOLUTION, MCF values were computed. The mean sample fraction for each mutation was computed as

$$MSF = \frac{VAF \times \left( \left( CN_{\mathrm{mutant}} + CN_{\mathrm{wildtype}} \right) \times TCF + 2(1 - TCF) \right)}{M} \qquad (8)$$

where $CN_{\mathrm{mutant}}$ is the copy number of the mutant allele, $CN_{\mathrm{wildtype}}$ is the copy number of the wildtype allele, M is the number of mutated alleles, TCF is the tumor cell fraction, and the VAF is the variant allele frequency of the mutation. For the data set, the TCF was determined by computing the maximum closest to VAF 0.5 for the density plot of the VAF. Multiplying this by 2 gives the TCF. Subsequently, the MCF was computed as

$$MCF = \frac{MSF}{TCF} \qquad (9)$$

A segment file for each tumor was constructed which was used as an input to DEVOLUTION. The MCF computations were also performed for all events that passed quality control i.e., also the private mutations excluded in the MAGOS analysis. Phylogenetic trees were thus reconstructed based on the MAGOS cluster nesting, the DEVOLUTION output using the mutations feasible to cluster with MAGOS as well as the DEVOLUTION output on all mutations that passed QC (Supplementary Figs. 14–17).

**Statistics and reproducibility**. For phylogenetic tree characteristics, significance was tested using the Mann–Whitney U-test (two-tailed). The stem lengths, total branch lengths, and number of branches between individual patients are assumed to be independent of each other and not normal distributed. In the violin plots in Figs. 3a–c, 4e, and 5n, o and Supplementary Fig. 13 box plots illustrate the interquartile range and the red dot the median value.

All simulation points were repeated at least three times and the standard deviation are illustrated with bars in Fig. 4c, d.

**Reporting summary**. Further information on research design is available in the Nature Research Reporting Summary linked to this article.

## Data availability

All data generated or analyzed during this study are included in this article as Supplementary Data 1–6. The neuroblastoma, Wilms tumor, and rhabdomyosarcoma data are also part of a previous study[17]. WES data from the TRACERx tumors were extracted from cBioPortal: http://www.cbioportal.org/study/summary?id=nsclc_tracerx_2017 copy number summaries were available in the supplementary tables of the corresponding study[31].

## Code availability

The code is freely available and relies on R 4.0.2 or later. Setup instructions and dependencies can be found on github. https://github.com/NatalieKAndersson/DEVOLUTION.

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

## Acknowledgements

We would like to thank the Swegene Centre for Integrative Biology at Lund University (SCIBLU) for assistance.

## Author contributions

N.A. and D.G. conceived and designed the project. N.A., D.G., and S.C. developed the methodology. A.V., J.K., and D.G. did tumor biopsy data acquisition. N.A. and D.G. analyzed and interpreted the data. N.A., D.G., S.C., A.V., and J.K. contributed towards the manuscript.

## Funding

## Competing interests

The authors declare no competing interests.
