## [Transparent Peer Review File · Communications Biology]

Reviewers' comments:

Reviewer #1 (Remarks to the Author):

In the manuscript by Andersson et al. the authors present DEVOLUTION, a new algorithm and tool for subclonal inference and phylogenetic reconstruction from either copy number or sequencing data, or both, which allows for integrating information from multiple biopsies. The algorithm utilizes information on the position and type of each alteration, as well as the proportion of cells harboring the alteration, to infer an event matrix that describes the distribution of each alteration in different subclones. The phylogenetic distances between subclones is then inferred from the event matrix, to reconstruct phylogenetic trees, and visualize the size of clusters in different samples. An important strength of the algorithm is the ability utilize only CNAs as input, which are more frequently associated with pediatric cancers. Therefore, the algorithm was applied to 253 tumor regions from 56 pediatric cancers, to reconstruct event matrices and phylogenetic trees (for 50 of the 56 cancers), showing overall high agreement between maximum likelihood and parsimony reconstructed trees. Finally, the authors evaluate the robustness of their algorithm using simulated bulk data.

The approach presents important advantages over existing techniques, by evaluating combined information from multiple biopsies, and considering both point mutations and copy number alterations data. Therefore, it could be a significant addition to existing strategies for tumor phylogenetic reconstruction. However, the input data and code must be provided in full to allow evaluation of DEVOLUTION and reproducibility of the results. In addition, the authors should discuss potential limitations and use cases for their tool, and compare the results and insights to those obtained with existing techniques.

Specific comments:

1. The pediatric tumor samples that were used throughout this study are not available or incompletely described in the manuscript, seriously impeding reproducibility of the results. The authors write "DEVOLUTION was applied to a previously reported dataset of 56 pediatric cancers and phylogenetic trees were generated based on copy number data for 22 neuroblastomas, 20 Wilms tumors and 8 rhabdomyosarcomas comprising a total of 253 biopsies" What is the previously reported dataset? It is missing a reference or accession, and the data itself is not available from the manuscript. The authors should include a methods section describing the data with references and accessions, and if possible, also provide the data as it is used as input for DEVOLUTION.
2. The code for DEVOLUTION is not provided. The github directory only contains a README and .png files. The R code and example input data (which is currently not available in the github page) must be provided for the evaluation of this tool, and reproducibility of the results.
3. The authors write "Evaluating the software using high-throughput SNP-array data from multitemporal and spatial regions from 56 pediatric tumors produced phylogenetic trees that were well in concordance with prior knowledge of how chromosomal aberrations occur in cancer cells." However, this analysis is incomplete. What are the events that are in agreement with known biological mechanisms? In how many of the tumors? any know early, late or metastatic events that are correctly inferred? How are these prediction in comparison to those made by existing tools, when applied to the same data? A comprehensive evaluation and comparison are necessary.
4. Table 1 is very helpful for evaluation DEVOLUTION in comparison with existing tools. However, it is not clear from the manuscript what are the possible limitations of this approach. A discussion of the limitations of DEVOLUTION for different objectives and with different types of input is necessary.
5. In addition, it would be helpful if the authors would discuss utilization of DEVOLUTION for other cancer types, and mention other types of cancer where CNA-based phylogeny may be important and

useful.

6. Could the authors try to explain why they observe ML/MP discrepancy in 50% of the Wilms tumors, even when excluding cases of whole chromosomes?

7. Related to that, Do the authors exclude only whole chromosomes, or also chromosomal arm events? As many cancer specific aneuploidies occur in the arm level, and not in the whole chromosome level.

Reviewer #2 (Remarks to the Author):

Andersson et al. develop a new method for phylogenetic reconstruction of tumor clonal lineages specifically designed for tracking copy number variations in aneuploid cancers. While there is now a crowded field of methods for tumor phylogenetics, the paper does a good job of summarizing the literature, identifying an important area in need of development, and explaining how their method can fill that need. The authors make a good case specifically for methods better suited to aneuploid genomes. They also provide a reasonable argument for developing such a method for use with bulk data in need of deconvolution, even though much of the interest in the field has shifted to single-cell. The developed method appears sensible and provides some useful features lacking in prior work. The authors provide a good demonstration of accuracy on simulated data for a range of parameter assumptions. They then apply the method to a diverse cohort of pediatric cancers, showing effectiveness in making at least biologically plausible reconstructions and leading to some interesting conclusions about the cancers and about the practice of tumor phylogenetics. That said, the paper does raise some concerns, largely concerning questions about the methodology and the degree to which the method innovates beyond and improves on prior approaches, discussed below.

1. The paper generally does a very good job of summarizing the past literature, at least with respect to the most widely used and cited methods, and explaining how the present work is differentiated from them but could use more focus on alternative methods for examining aneuploid genomes. The argument is fair that almost all methods either ignore aneuploidy altogether or treat it as an afterthought to studying evolution by single nucleotide variations. That has been changing, though. Some of the oldest methods in the field were focused specifically on CNAs (e.g., MEDICC) and many recent methods allow for copy number variation or even focus primarily on it (e.g., HATCHET). The paper would benefit from some brief consideration of the literature of methods for CNA evolution specifically, again highlighting what is distinctive about the present method and what it teaches one about the biology that could not have been learned from prior methods.

2. Related to that, I believe the work would benefit from some more direct head-to-head comparison with alternatives, even if they do not use exactly the same data. It would seem likely that some methods would be able to work with at least a subset of the data used by DEVOLUTION, and give trees on which one could make head-to-head comparisons of accuracy on simulated data by the various measures considered. The paper might also consider how outputs on the real data would differ between DEVOLUTION versus a method not specifically designed for aneuploidy, to better make the case that one learns new insights into the biology because of the improvements in the present work.

3. My biggest concern about the paper is that some of the methods are unclear to me from the text, which makes it difficult to judge significance and technical soundness of the work as a contribution to tumor phylogenetics. The general idea behind the method is standard for the field, essentially clustering mutations by frequency to identify clones. It is a sound if not very innovative approach in itself. The actual description of the algorithm is fairly informal, though and would benefit from a more precise formal description of the algorithm, such as pseudocode. In addition, aspects of the method,

such as the selection of the number of clones, are described as manually implemented heuristics rather than precise algorithms and would be better formalized, integrated with the code, and tested.

4. The actual phylogenetics relies on a third-party implementation of standard methods (maximum likelihood or maximum parsimony) although. There, too, a more precise description of the assumed models would be useful. This includes a description of the precise probability function behind the likelihood model as well as some clarification of what is being counted in the Hamming distance computations (variant bases, breakpoints, CNAs, etc.). In addition, it would be useful to see a justification of these models on the biology, to argue that they describe a reasonable mathematical model of clonal evolution in cancers as we understand it. Much of the work in tumor phylogenetics is on the question of how to improve phylogenetic models and algorithms to better capture the biology of tumor evolution, so it does raise concern that this work is in some ways seemingly a step backward from the state of the art in using standard generic phylogeny algorithms. (Although it is interesting that the solution is often insensitive to the algorithm).

5. Another particular point of concern on methods is the assumption that the method takes as input a set of mutated clone fractions (MCF). I am not sure if this is exactly the same as what is sometimes called the cancer cell fraction (CCF) in other works in this domain, but it appears to be the same idea or at least very similar. The problem is that these values are notoriously difficult to estimate accurately in non-diploid genomes from raw read counts or the equivalent since doing so requires separating allele frequency contributions due to clonal composition from those due to copy number. I would normally expect one of the primary contributions of a new method in this space to be a better way of solving this particular problem of estimating MCF or CCF. The paper appears to allow that MCFs can be derived by preprocessing with a third-party method, but that answer is problematic since this derivation is normally done in the course of clonal deconvolution and phylogenetics and its answer depends on the solution to those problems. Perhaps I misunderstand here, in which case all that is needed might be a clarification of what is done and why it is technically sound.

6. The application to pediatric cancers provides some additional validation and yields some interesting insights, although the cohort size may be too small to read too much into the specific findings. The general consistency of ML and MP phylogenies is a nice observation and the work is useful in providing some insight into recurring reasons inconsistencies sometimes occur. It would be worth paying more attention to whether any of the main observations, such as higher branch length of RMS tumors, are statistically significant and distinguish those findings that are anecdotal from those that are statistically sound. The cohort is sufficiently small and heterogeneous, though, that it is understandable many questions will not be sufficiently powered.

Although the paper is otherwise nicely written, I did note a few minor errors:

7. line 109: "should never be occur" should be "should never occur"
8. line 274: "genetic alterations that seems" should be "genetic alterations that seem"
9. line 280: "in well in concordance to" should be "well in concordance with"

Reviewer #3 (Remarks to the Author):

Andersson et al. present DEVOLUTION, a new tool to reconstruct clonal structure and phylogenetic trees from multiregional cancer genomics data. This work is an interesting contribution in the context of the growing field of clonal deconvolution analysis. One novelty of this tool is that it allows phylogenetic tree reconstruction solely based on copy number alterations (CNA), which are common features of many cancer types. I have the following comments for the authors to address:

- 1) Case examples presented to illustrate how the tool works were taken from a dataset of pediatric

cancers (Fig 2). The cases discussed are unusually data-rich, with five or more samples per patient, often associated with multiple sites and clinical progression. It is not clear what the utility is for this tool in more common scenarios, for instance when only one (baseline) or two (pre- vs post-treatment) samples per patient are available.

2) How does phylogenetic tree reconstruction based on SNP-array differ from one using WES? This comparison would be useful to assess the robustness of the approach.

3) DEVOLUTION provides optimal phylogenetic trees based on two criteria, maximum likelihood (ML) and parsimony (MP), which in turn are shown to often agree and have minor differences. However, it'd be important to also provide information on suboptimal solutions. One might have a scenario with two or more solutions of similar likelihood, so that the optimal one might be unstable. An overall score or measure of robustness/stability would be desirable, as well as the ability to inspect and analyze alternative solutions.

Response to Reviewer Comments

Below we respond point-by-point to the reviewer comments. The comments are copied verbatim but bold type transformation has been added by us to clarify the most important points. Our response is detailed in red type.

Reviewer 1:

1. The pediatric tumor samples that were used throughout this study are not available or incompletely described in the manuscript, seriously impeding reproducibility of the results. The authors write “DEVOLUTION was applied to a previously reported dataset of 56 pediatric cancers and phylogenetic trees were generated based on copy number data for 22 neuroblastomas, 20 Wilms tumors and 8 rhabdomyosarcomas comprising a total of 253 biopsies” What is the previously reported dataset? It is missing a reference or accession, and the data itself is not available from the manuscript. The authors should include a methods section describing the data with references and accessions, and if possible, also provide the data as it is used as input for DEVOLUTION.

The correct reference has now been added, with a first occurrence at line 79. The complete data set was also available through the supplementary files provided with the first version of the manuscript.

2. The code for DEVOLUTION is not provided. The github directory only contains a README and .png files. The R code and example input data (which is currently not available in the github page) must be provided for the evaluation of this tool, and reproducibility of the results.

The complete extensively commented code has been added to github.

<https://github.com/NatalieKAndersson/DEVOLUTION/tree/master>

3. The authors write “Evaluating the software using high-throughput SNP-array data from multitemporal and spatial regions from 56 pediatric tumors produced phylogenetic trees that were well in concordance with prior knowledge of how chromosomal aberrations occur in cancer cells.” However, this analysis is incomplete. What are the events that are in agreement with known biological mechanisms? In how many of the tumors? any know early, late or metastatic events that are correctly inferred? **How are these prediction in comparison to those made by existing tools, when applied to the same data? A comprehensive evaluation and comparison are necessary.**

What are the events that are in agreement with known biological mechanisms? In how many of the tumors? any know early, late or metastatic events that are correctly inferred? In our previous article we evaluated early vs late events across NB, WT and RMS (“Extensive clonal branching shapes the evolutionary relationship of high-risk pediatric tumors” published in Cancer Research 2020).

A motivation has been added to the section “*Validation on pediatric tumors confirms concordance to established scenarios*”.

- **Neuroblastoma:** Known early events such as
 - **MYCN-amplification:** Was declared to the stem in 7/7 tumors in which it was present.
 - **Whole chromosome 17 gain:** Declared to the stem in 8/9 tumors. In the discrepant case (NB10) merely two biopsies were present of which +17 was 100 % in one of them.
 - **17q gain:** Declared to the stem in 8/13 tumors. In all the remaining cases it was present in all biopsies available from the patient, although not ≥ 90 % in all of them,

nonetheless indicating that this genetic alteration consistently appears early in the evolution of these tumors.

- **WT: 11p cni** in WT was declared to the stem in 7/9 cases. The remaining cases were one case in a tumor only including 3 genetic alterations of which the 11p alteration made up 60 % of one of the biopsies. The other case was a complex case with 4 different types of cni:s in 11p that were early in the branching of the phylogeny.
- **Rhabdomyosarcoma: PAX3/FOXO1** as well as **PAX7/FOXO1** fusion genes in RMS was declared to the stem in all instances.

How are these predictions in comparison to those made by existing tools, when applied to the same data? A comprehensive evaluation and comparison are necessary. An extensive comparison has now been made to MAGOS, a recent software tool which has been shown to outperform older tools such as PyClone and SciClone. The comparison was made using the external TRACERx-data set comprising whole exome sequencing data from non-small cell lung cancers.

MAGOS, and similar tools, can only analyze genetic alterations present in all biopsies. Therefore, phylogenetic trees were produced using DEVOLUTION on merely the mutations possible to include in the MAGOS-analysis as well as phylogenetic trees based on all genetic alterations that passed quality control i.e. also including genetic alteration present in a subset of biopsies, fully displaying the heterogeneity of the tumor. MAGOS clusters genetic alterations that show similar patterns in VAF:s across samples. To produce a phylogeny, or obtain the subclones, nesting of the clusters still must be made. Here we did a manual nesting of the MAGOS-clusters based on their sizes across samples. The resulting three sets of phylogenies for each tumor was subsequently compared.

The result of this analysis can be found under the section “*Using an external dataset for benchmarking*” as well as Figure 5 c-f, Supplementary Figure 10 and Supplementary File 6.

4. Table 1 is very helpful for evaluation DEVOLUTION in comparison with existing tools. However, it is not clear from the manuscript what are the possible **limitations** of this approach. A discussion of the limitations of DEVOLUTION for different objectives and with different types of input is necessary.

We have added a section in the Discussion on the limitations of DEVOLUTION starting at line 323.

5. In addition, it would be helpful if the authors would **discuss** utilization of DEVOLUTION for other cancer types, and mention other types of cancer where CNA-based phylogeny may be important and useful.

Other types of cancer where CNA-based phylogeny may be important and useful: In the Introduction (starting at line 62) we have added extended the paragraph “...cancer types where aneuploidy is a common feature such as high-grade adult carcinomas, high-grade brain tumors and many childhood cancers.” We have also here added a reference to the paper “Pan-cancer whole-genome analyses of metastatic solid tumors” that included 2520 paired tumor and metastasis biopsies from 20 different tumor types. They found extensive copy number aberrations across all cancer types analyzed. In addition, 55.9 % of the samples displayed Whole Genome Doublings ranging in prevalence from 15 % of biopsies in CNS tumors to 80 % in esophageal tumors.

We have also added an analysis of non-small cell lung cancer (NSCLC) based on the TRACERx data set to address the utility for adult cancer. “*Using an external dataset for benchmarking*” as well as **Figure 5 c-f, Supplementary Figure 10 and Supplementary File 6.**

6. Could the authors try to explain why they observe ML/MP discrepancy in 50% of the Wilms tumors, even when excluding cases of whole chromosomes?

I presume the Referee refer to RMS, where there was indeed such discrepancy. This has already been expanded upon in the Results section “Contradictions are rarely seen in the phylogenetic trees” starting at line 184. We found that the RMS tumors seemingly have very complex genomes with a significantly higher branch- and stem length than was found in NB and WT (Figure 3b). In addition, most of the identified genetic alterations were > 50 %, hence restricting the possible solutions of the nesting of the genetic alterations across samples, resulting in a few instances of parallel evolution in the phylogenetic trees in 4 of the cases. We have also added the sentence “These two aspects explain the residual difference between the MP- and ML-trees” for further clarification to the reader.

7. Related to that, Do the authors exclude only whole chromosomes, or also chromosomal arm events? As many cancer specific aneuploidies occur in the arm level, and not in the whole chromosome level.

Good observation. Yes, we excluded chromosome arm events as well. In the cases in this study, no such events were the ones causing the contradictions. We have although clarified this in the text (line 180-181) “...and chromosome arms”, to prevent confusion regarding this.

Reviewer 2:

1. The paper generally does a very good job of summarizing the past literature, at least with respect to the most widely used and cited methods, and explaining how the present work is differentiated from them but could use more **focus on alternative methods for examining aneuploid genomes**. The argument is fair that almost all methods either ignore aneuploidy altogether or treat it as an afterthought to studying evolution by single nucleotide variations. That has been changing, though. Some of the oldest methods in the field were focused specifically on CNAs (e.g., **MEDICC**) and many recent methods allow for copy number variation or even focus primarily on it (e.g., **HATCHET**). The paper would benefit from some **brief consideration of the literature of methods for CNA evolution specifically**, again highlighting what is distinctive about the present method and what it teaches one about the biology that could not have been learned from prior methods.

We have moved the summarization of past literature to the Discussion. In addition, we have revised and extended the focus on methods for CNA evolution, in line with your suggestions. We have also clarified the distinction to DEVOLUTION. This section can be found in the Discussion ranging from line 340-375.

2. Related to that, I believe the work would benefit from some more **direct head-to-head comparison** with alternatives, even if they do not use exactly the same data. It would seem likely that some methods would be able to work with **at least a subset of the data** used by DEVOLUTION, and give trees on which one could make head-to-head comparisons of accuracy on simulated data by the various measures considered. The paper might also consider how outputs on the real data would differ between DEVOLUTION **versus a method not specifically designed for aneuploidy**, to better make the case that one learns new insights into the biology because of the improvements in the present work.

Direct head-to-head comparison with alternatives: An extensive comparison has now been made to MAGOS, a recent software tool which has been shown to outperform older tools such as PyClone and SciClone. The comparison was made using the external TRACERx-data set comprising whole exome sequencing data from non-small cell lung cancers.

MAGOS, and similar tools, can only analyze genetic alterations present in all biopsies. Therefore, phylogenetic trees were produced using DEVOLUTION on merely the mutations possible to include in the MAGOS-analysis as well as phylogenetic trees based on all genetic alterations that passed quality control i.e. also including genetic alteration present in a subset of biopsies, fully displaying the

heterogeneity of the tumor. MAGOS clusters genetic alterations that show similar patterns in VAF:s across samples. To produce a phylogeny, or obtain the subclones, nesting of the clusters still must be made. Here we did a manual nesting of the clusters based on their sizes across samples. The resulting three sets of phylogenies for each tumor was subsequently compared.

The result of this analysis can be found under the section “*Using an external dataset for benchmarking*” as well as Figure 5 c-f, Supplementary Figure 10 and Supplementary File 6.

Aneuploidy: In supplementary Figure 8 and Figure 5 a-b we have also made extensive evaluations of DEVOLUTION on data with sequencing data alone, SNP-array data alone and when the two data sets were combined for 18 pediatric tumors where both types of data were available. Here we could see that additional information about the tumor evolution is obtained when combining the data sets. NB16 illustrates a tumor that is near diploid. This results in a very rudimentary phylogenetic tree based on copy number data alone. Addition of data from other methods such as WES or TDS provides additional information on the evolution of the patient’s tumor. But we can also see the importance of including copy number aberrations in the analysis of pediatric tumors. For the same tumor, NB16, we find stem events that are well established predictors of poor prognosis and treatment failure in neuroblastoma, such as a MYCN-amplification and 1p deletion and a subclonal 17q gain, that would not be acknowledged if merely including sequencing data.

3. My biggest concern about the paper is that some of the methods are unclear to me from the text, which makes it difficult to judge significance and technical soundness of the work as a contribution to tumor phylogenetics. The general idea behind the method is standard for the field, essentially **clustering mutations by frequency** to identify clones. It is a sound if not very innovative approach in itself. The actual description of the algorithm is **fairly informal**, though and would benefit from a more precise formal description of the algorithm, such as **pseudocode**. In addition, aspects of the method, such as the **selection of the number of clones**, are described as manually implemented heuristics rather than precise algorithms and would be better formalized, integrated with the code, and tested.

Clustering: Clustering of mutations is in our algorithm merely a first step that is made to aid the computation further on and is not the main task for DEVOLUTION. Just clustering events by frequency does not infer the subclones, merely the events that are characteristic for a certain subclone. The clonal nesting must still be made to infer which subclones of cells are present within the tumor, which is the main task which the DEVOLUTION algorithm is responsible of.

This has previously been clarified at line 475: “Note that for DEVOLUTION the clustering is only used to reduce the computational complexity for the upcoming subclonal deconvolution algorithm.”

It is now further clarified in the discussion at line 321: “The purpose of the ad hoc clustering mainly is to reduce the computational complexity and not to find clones and is not to be confused with the clustering used in dedicated clustering algorithms for clonal determination. The choice of clustering method can also easily be changed by the user.”.

Selection of number of clones: There is no manual selection of number of clones. The preclustering made does not need a user to choose the number of clusters. The algorithm itself determines the optimal number of clusters. Additionally, the number of subclones in the end can differ from this amount. For example, if you have a case where you have 5 events that are 100 %, 3 events that are 80 % and 1 event that is 40 %. In this case, the clustering will produce three clusters, without the user having to declare the number of clusters on beforehand. Then the algorithm will nest these clusters to decipher which subclones are present in the tumor, which in this case would be 3 different subclones.

We have reformulated the method section “Clustering of genetic alterations incorporating information from multiregional and temporal sampling from the same patient”.

Pseudocode: It would surprise us if this is needed since the code is already extensively commented and should be relatively easy to follow.

Fairly informal, though and would benefit from a more precise formal description of the algorithm:

The method provides the major steps and ideas of all the larger subalgorithms in the complete algorithm. This overview description in combination with the extensively commented code and example files on github, should be enough for the user to follow what the algorithm is doing.

4. The actual phylogenetics relies on a third-party implementation of standard methods (maximum likelihood or maximum parsimony) although. There, too, a more precise description of the assumed models would be useful. This includes a description of the **precise probability function** behind the likelihood model as well as some clarification of **what is being counted in the Hamming distance computations** (variant bases, breakpoints, CNAs, etc.). In addition, it would be useful to see a **justification of these models on the biology**, to argue that they describe a reasonable mathematical model of clonal evolution in cancers as we understand it. Much of the work in tumor phylogenetics is on the question of how to improve phylogenetic models and algorithms to better capture the biology of tumor evolution, so it does raise concern that this work is in some ways seemingly a step backward from the state of the art in using standard generic phylogeny algorithms. (Although it is interesting that the solution is often insensitive to the algorithm).

Precise probability function behind the likelihood model: In this study we used the Jukes Cantor model as substitution model for the optim.pml algorithm. Hence assuming equal transition rates and equilibrium frequencies for all states. OptEdge was set to TRUE. This has been clarified in line 586. Please let me know if this was not what you were referring to.

Hamming distance: We have clarified in the text what is being counted in the Hamming distance computations at line 558-559.

“Here, the Hamming distance between the subclones in the event matrix was used to assess a distance matrix displaying the genetic distance between each of the subclones. It computes the distance between two vectors by adding all positions in which they differ from one another, in this case the number of genetic alterations or positions in the event matrix the entities differ from one another, resulting in branch length in units of number of aberrations.”

ML and MP: The point of the DEVOLUTION algorithm is to infer subclones and construct an event matrix illustrating which genetic alterations are present in each such identified subclone. The last part is then to construct phylogenetic trees based on this event matrix, which here is exemplified with ML and MP. If the user wants to another method could be used. We here chose to use ML and MP since these are well-established methods that generally work well for small data sets.

Bayesian methods using variants of Markov chain Monte Carlo sampling have a much higher computational cost. You must also determine a prior distribution, which can be hard, and you will also make some subjective marks on the result. You could although obtain multiple plausible trees, while ML often only gives you one tree.

We have now clarified the choice of ML and MP in the text in the section reconstruction of phylogenetic trees around line 568-575.

5. Another particular point of concern on methods is the assumption that the method takes as input a set of mutated clone fractions (MCF). I am not sure if this is exactly the same as what is sometimes called the cancer cell fraction (CCF) in other works in this domain, but it appears to be the same idea or at least very similar. The problem is that these values are **notoriously difficult to estimate** accurately in non-diploid genomes from raw read counts or the equivalent since doing so requires separating allele frequency contributions due to clonal composition from those due to copy number. I would normally expect one of the primary contributions of a new method in this space to be a better

way of solving this particular problem of estimating MCF or CCF. The paper appears to allow that MCFs can be derived by preprocessing with a third-party method, but that answer is problematic since this derivation is normally done in the course of clonal deconvolution and phylogenetics and its answer depends on the solution to those problems. Perhaps I misunderstand here, in which case all that is needed might be a clarification of what is done and why it is technically sound.

Yes, MCF and CCF refers to the same thing. The proportion of the cancer cells in a biopsy that have a certain genetic alteration. Using the MCF as input allows analysis of genetic alterations from different methods in unison.

A tool, currently in review, named DeCiFer reliably infers CCFs from SNVs (<https://github.com/raphael-group/decifer>). Also, SVClone published in 2020 in Nature Communications (<https://www.nature.com/articles/s41467-020-14351-8>), computes CCF for structural variants.

CCFs from VAFs and copy number aberrations can be computed by methods previously published (<https://www.nejm.org/doi/full/10.1056/NEJMoa1616288>, <https://www.nature.com/articles/s41588-018-0131-y>).

DEVOLUTION on the other hand, mainly provides a way to thoroughly nest clusters of genetic alterations based on multiregional and multispatial information. Through simulations we also show that if the CCFs are reliable and there are multiple biopsies available, reliable phylogenetic trees are obtained.

These issues have now been clarified in the revised Discussion starting at line 323.

“While DEVOLUTION enables further analysis of heterogenous multiregional and temporal tumorigenic data there are still venues for improvements. Firstly, using the MCF as an input for the algorithm stresses the need of a robust pre-analysis of the data set. The inferred subclones identified could be affected by the choice of method for computation of the MCF-values. Methods to compute these has been discussed extensively elsewhere and novel methods are being developed”. See the section in the manuscript for references.

6. The application to pediatric cancers is provides some additional validation and yields some interesting insights, although the cohort size may be too small to read too much into the specific findings. The general consistency of ML and MP phylogenies is a nice observation and the work is useful in providing some insight into recurring reasons inconsistencies sometimes occur. It would be worth paying more attention to whether any of the main observations, such as higher branch length of RMS tumors, **are statistically significant** and distinguish those findings that are anecdotal from those that are statistically sound. The cohort is sufficiently small and heterogeneous, though, that it is understandable many questions will not be sufficiently powered.

Pediatric tumors are rare compared to adult cancers, which is why the data set here is seemingly small compared to studies of adult cancers.

Already in the first version of the manuscript, p-values were displayed in Figure 3a for the cases where the difference was significant (p-value < 0.05). P-values were computed using a two-sided Mann-Whitney U-test. We could see that the RMS tumors had a significantly higher total branch length and stem length than NB and WT, indicating a more complex genomic profile. This has now been clarified further in the text in the section “Contradictions are rarely seen in the phylogenetic tree.” starting at line 184.

Although the paper is otherwise nicely written, I did note a few minor errors:

7. line 109: “should never be occur” should be “should never occur”

8. line 274: “genetic alterations that seems” should be “genetic alterations that seem”
9. line 280: “in well in concordance to” should be “well in concordance with”

These errors have been corrected in line 111, line 383 and line 390.

Reviewer 3:

1) Case examples presented to illustrate how the tool works were taken from a dataset of pediatric cancers (Fig 2). The cases discussed are unusually data-rich, with five or more samples per patient, often associated with multiple sites and clinical progression. It is not clear what the utility is for this tool in more common scenarios, for instance when only one (baseline) or two (pre- vs post-treatment) samples per patient are available.

This has already been evaluated using the simulated data at the end of the Results section. There we could see how the algorithm could handle cases where we had 1 to 10 biopsies. This has now been clarified further in Results starting at line 208, with reference to specific figure panels (Fig. 4b and c). In addition, the new comparison with SNP array vs WES and DEVOLUTION vs MAGOS (e.g. Fig. 5) includes cases with fewer biopsy samples.

2) How does phylogenetic tree reconstruction based on SNP-array differ from one using WES? This comparison would be useful to assess the robustness of the approach.

We have now added an extensive comparison. This can be found in the result section named “Unifying sequencing and SNP-array data reveals additional evolutionary pathways”, Figure 5 a-b as well as Supplementary Figures 8 and 9. Here we produced phylogenetic trees using sequencing data alone, SNP-array data alone and with the two data sets combined for 8 NB, 9 WT and 3 RMS for which both data types were available.

3) DEVOLUTION provides optimal phylogenetic trees based on two criteria, maximum likelihood (ML) and parsimony (MP), which in turn are shown to often agree and have minor differences. However, it'd be important to also provide information on **suboptimal solutions**. One might have a scenario with two or more solutions of similar likelihood, so that the optimal one might be unstable. An overall score or measure of robustness/stability would be desirable, as well as the ability to inspect and analyze alternative solutions.

This is a very good point. This is although something that the user can investigate outside the DEVOLUTION algorithm when doing analysis with phangorn (or another software if that is of preference). We therefore find that adding this capability to DEVOLUTION is of limited value as it extensively overlaps with well-established tools.

Reviewers' comments:

Reviewer #1 (Remarks to the Author):

I thank the authors for their responses, my concerns have been fully addressed in the revised manuscript.

Reviewer #2 (Remarks to the Author):

I have read the revisions and response to reviewers and find them largely responsive to the prior concerns. The paper does a much better job of summarizing the prior literature and describing the novel advantage of the present method. The authors have also added extensive comparisons with a related method and across various data assumptions, making a good case for the novel advance of the present work. The model and methods are clarified in several important respects, making the specific contribution clearer. I believe the paper is much improved and makes a stronger case that this work is a methodological advance in understanding clonal evolution and heterogeneity in aneuploid cancers, in addition to providing some new insights into the specific pediatric cancers examined.

There are a few points on which I find the responses not entirely satisfactory. I do still maintain that the methods, at the level of algorithms, are less clear and precise than is generally the standard at least for the method developer community. While well commented code is appreciated, it is a different thing than a clear and unambiguous specification of the algorithm. The clarifications also do reinforce the prior impression that the advance is relatively modest as methodology per se. It is fair enough to say that phylogenetics is not part of the contribution. I remain more concerned that computing MCF is not part of the contribution, since that is a difficult problem to get right on aneuploid cancers, in part because the solutions to MCF, deconvolution, and phylogenetics all depend on each other and are not easily separable. I can accept, though, that the paper makes a reasonable empirical case that it does improve on prior methods to treat MCF and phylogenetics and pre- and post-processing steps that can be handled adequately by third-party methods.

Reviewer #3 (Remarks to the Author):

The authors satisfactorily addressed the first two questions from my report. However, I feel the third one has not been satisfactorily responded. I believe questions of stability of optimal solutions, how to assess suboptimal solutions, and how to integrate them into the analysis workflow, are of great importance for readers and users of their tool. I encourage them to provide a case example, guidance and discussion of this topic.

Reviewers' comments

Below, reviewer comments to the revised version are copied verbatim with bold text formatting done by the authors to highlight the points of criticism.

Reviewer #1 (Remarks to the Author):

I thank the authors for their responses, my concerns have been fully addressed in the revised manuscript.

Reviewer #2 (Remarks to the Author):

I have read the revisions and response to reviewers and find them largely responsive to the prior concerns. The paper does a much better job of summarizing the prior literature and describing the novel advantage of the present method. The authors have also added extensive comparisons with a related method and across various data assumptions, making a good case for the novel advance of the present work. The model and methods are clarified in several important respects, making the specific contribution clearer. I believe the paper is much improved and makes a stronger case that this work is a methodological advance in understanding clonal evolution and heterogeneity in aneuploid cancers, in addition to providing some new insights into the specific pediatric cancers examined.

There are a few points on which I find the responses not entirely satisfactory. I do still maintain that the methods, at the level of algorithms, are **less clear and precise than is generally the standard** at least for the method developer community. While well commented code is appreciated, it is a different thing than **a clear and unambiguous specification of the algorithm**. The clarifications also do reinforce the prior impression that the advance is relatively modest as methodology per se. It is fair enough to say that phylogenetics is not part of the contribution. **I remain more concerned that computing MCF is not part of the contribution, since that is a difficult problem to get right on aneuploid cancers, in part because the solutions to MCF, deconvolution, and phylogenetics all depend on each other and are not easily separable.** I can accept, though, that the paper makes a reasonable empirical case that it does improve on prior methods to treat MCF and phylogenetics and pre- and post-processing steps that can be handled adequately by third-party methods.

Author response on specification of the algorithm:

Thank you for these comments and criticisms. We have addressed the concerns regarding the unsatisfactory algorithm description. A flow chart encompassing the functions within the code is now attached as **S. Figure 12** as a complement to the commented code. We have also in text composed a comprehensive description of the algorithm, now enclosed as Supplementary Information. In this document, we explain what each segment of the original code does. We hope this level of detail in the revised version is now satisfactory.

Author response on the use of MCF as input parameter:

We completely agree that computing MCF (the proportion of cancer cells which a certain clone encompasses/tumor cell fraction) is not trivial. However, in our opinion, MCF is the most straightforward way to integrate data on sequence mutations, copy number alterations, and copy number neutral allelic imbalances. These genomic changes are often present together in cancer cells, and any tool for clonal deconvolution that fails to integrate data from all three will thus risk being suboptimal.

The main reason we did not integrate MCF calculations in the current toolkit is that we have in parallel developed a specific tool (CRUST) for subclonal deconvolution for sequence mutations, with integrated normalization for tumor cell purity and allelic composition. This tool is described in a preprint found at

doi.org/10.1101/2020.11.11.376467

and the paper is currently in press in *Briefings in Bioinformatics*. CRUST is meant to be a complement to the current paper as it helps calculating MCFs from variant allele frequencies. A more rudimentary way of calculating MCF from sequence data for integration with copy number data has also been described by us in Karlsson et al. 2018 (Nat Genet PMID: 29867221), a paper we already cite in conjunction with first mentioning MCF in the previous version of this paper (p. 4).

Regarding calculations of MCF for copy number alterations, this is considerably more straightforward. Calculations of MCF from B-allele frequencies and/or log2 ratios have been validated against experimental data (primarily fluorescence in situ hybridization) and then used by us and others for more than a decade. Seminal papers describing MCF calculations for copy number data and allelic imbalances are Staaf et al. 2008 (Genome Biol, PMID: 18796136), Gisselsson et al. 2010 (Proc Natl Acad Sci USA, PMID: 21059955), and Holmquist Mengelbier et al. 2015 (Nat Commun, PMID: 25625758).

To clarify that MCF is a useful and robust parameter for integrated clonal deconvolution, we have now added explanatory text in the Introduction (main text lines 54-59) and also added references to the most important papers as specified above.

Reviewer #3 (Remarks to the Author):

The authors satisfactorily addressed the first two questions from my report. However, I feel the third one has not been satisfactorily responded. I believe questions of stability of optimal solutions, how to **assess suboptimal solutions**, and how to integrate them into the analysis workflow, are of great importance for readers and users of their tool. I encourage them to provide a **case example, guidance and discussion** of this topic.

Thank you for your comment. We have now integrated an algorithm in the original code in which alternative solutions are assessed. The revised code searches for alternative solutions if present. If there are none, the user is informed. If there are alternative phylogenetic tree solutions that explains the observed data, the user is asked if an alternative solution should be shown, which is then done. The user is also provided with a matrix showing which of the subclones of the phylogeny are certain, meaning that the clusters within them only can be nested in a single way, and which are uncertain, indicating that clusters constituting those subclones could be nested in another way (**main text lines 235-244, S. Figure 11, S. Information, S. Figure 12**). There is also the possibility to color the tip labels of the phylogenetic trees accordingly for transparency. A case example can be seen in **S. Figure 11**. For the case where multiple solutions are possible, the user is also provided a matrix illustrating which subclones could possibly be present in the tumor and, in that case, in which samples. A discussion on this topic is also found in **S. Information** pages 14-15.

The updated code is uploaded to GitHub together with an excel sheet named "Multiple_solutions.xlsx" for which the same example as described in S. Figure 11 can be reproduced.

REVIEWERS' COMMENTS:

Reviewer #2 (Remarks to the Author):

I am satisfied with the reviewer's response. All of my pending concerns have been addressed.

Reviewer #3 (Remarks to the Author):

The authors have fully addressed my comments in the revised version of the manuscript. Thank you.